# Rethinking Point Cloud Data Augmentation: Topologically Consistent Deformation

**Jian Bi** [1]   **Qianliang Wu** [1]   **Xiang Li** [2]   **Shuo Chen** [3]   **Jianjun Qian** [1]   **Lei Luo** * [1]   **Jian Yang** * [1]

## Abstract

Data augmentation has been widely used in machine learning. Its main goal is to transform and expand the original data using various techniques, creating a more diverse and enriched training dataset. However, due to the disorder and irregularity of point clouds, existing methods struggle to enrich geometric diversity and maintain topological consistency, leading to imprecise point cloud understanding. In this paper, we propose SinPoint, a novel method designed to preserve the topological structure of the original point cloud through a homeomorphism. It utilizes the Sine function to generate smooth displacements. This simulates object deformations, thereby producing a rich diversity of samples. In addition, we propose a Markov chain Augmentation Process to further expand the data distribution by combining different basic transformations through a random process. Our extensive experiments demonstrate that our method consistently outperforms existing Mixup and Deformation methods on various benchmark point cloud datasets, improving performance for shape classification and part segmentation tasks. Specifically, when used with PointNet++ and DGCNN, our method achieves a state-of-the-art accuracy of 90.2 in shape classification with the real-world ScanObjectNN dataset. We release the code at https://github.com/CSBJian/SinPoint.

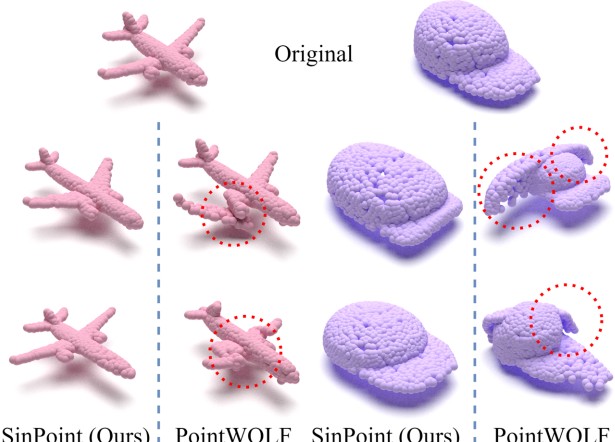

*Figure 1.* SinPoint ensures a smooth and natural deformation process, avoiding abrupt or unnatural changes. PointWOLF can lead to data distortion and significant semantic deviation.

## 1. Introduction

In machine learning and deep learning, both the quality and quantity of data are key factors in determining the performance of the model. However, in practical applications, obtaining a large amount of high-quality training data is often challenging, and the data itself may be biased, preventing the model from fully capturing the relevant features during training. These challenges have driven the development of Data Augmentation (DA) techniques. By applying various transformations to the original data, more diverse training samples can be generated, which helps mitigate the negative effects of insufficient data or distribution bias, thereby improving the model's generalization ability.

From a statistical perspective, the main purpose of data augmentation is to expand the distribution of training data. This allows the model to learn the diversity of the data and avoid over-reliance on specific patterns. Data augmentation applies transformations to the original data distribution, creating an extended one. The expectation is that the augmented distribution resembles the original, but with increased variance and a broader sample space. This approach enhances data diversity, reduces the model's dependence on specific

---

[1]PCA Lab, Key Lab of Intelligent Perception and Systems for High-Dimensional Information of Ministry of Education, School of Computer Science and Engineering, Nanjing University of Science and Technology, China [2]VCIP, CS, Nankai University, China [3]School of Intelligence Science and Technology, Nanjing University, China. Correspondence to: Lei Luo <cslluo@njust.edu.cn>, Jian Yang <csjyang@njust.edu.cn>.

*Proceedings of the $42^{nd}$ International Conference on Machine Learning*, Vancouver, Canada. PMLR 267, 2025. Copyright 2025 by the author(s).

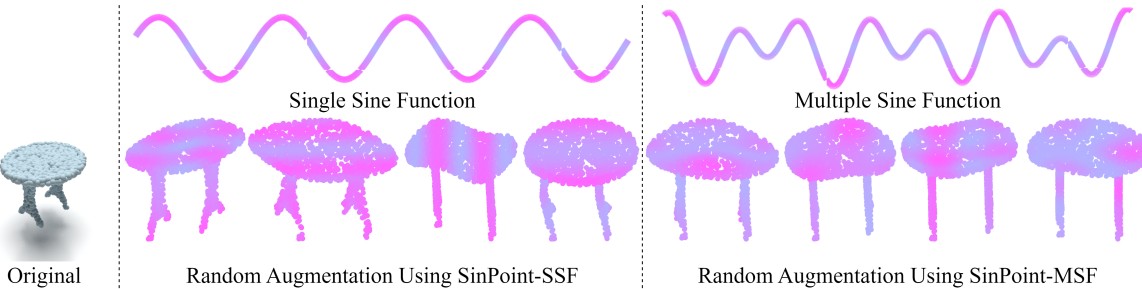

Figure 2. Some augmented samples produced by SinPoint-SSF (left) and SinPoint-MSF (right). The color is close to pink, the deformation is more significant. The data generated by the SinPoint-SSF object has only a single peak, while the samples from SinPoint-MSF have multiple distinct peaks.

samples, and helps it capture more general patterns.

In recent years, DA has shown significant success in handling image data. Various methods, such as Cutout (De-Vries & Taylor, 2017; Zhong et al., 2020), Mixup (Zhang et al., 2017), Cutmix (Yun et al., 2019), and other methods (Verma et al., 2019; Yang et al., 2022), have been utilized to augment images and enhance the robustness and generalizability of models. However, unlike regular images, point clouds are disordered and irregular, making it challenging to apply these DA methods directly to the point cloud data. Some existing point cloud DA methods only focus on a single type of operation, such as simple geometric transformations (rotation, scaling, and translation), data perturbations (adding noise and deleting points), or hybrid operations (simulated mixtures of images (Chen et al., 2020; Lee et al., 2022; 2021; Zhang et al., 2022; Ren et al., 2022; Wang et al., 2024)). While these methods may improve data diversity to some extent, they often overlook the point cloud's intrinsic structure and semantic details, resulting in a loss of topological consistency in the augmented point cloud. For instance, PointMixup (Chen et al., 2020), Point-CutMix (Zhang et al., 2022), and SageMix (Lee et al., 2022) all use different strategies to mix samples, but they do not consider the local structure of each sample. PointAugment (Li et al., 2020) relies on a learnable transformation matrix, making the outcome unpredictable. Similarly, PointWOLF (Kim et al., 2021) transforms local point clouds using a combination of strategies, which can lead to data distortion and significant semantic deviation, as shown in Figure 1.

In this paper, we analyze the essential reasons why data augmentation improves the model performance from a statistical point of view. On this basis, we propose a novel SinPoint transformation technique based on a homeomorphism (Derrick, 1973) to address these issues mentioned above. SinPoint aims to preserve the topological structure of the original point cloud by a homeomorphism and perturb the local structure using a Sine function to simulate the deformation of objects, thereby expanding the diversity of point clouds. We design two deformation strategies, as shown in Figure 2. One is to use a Single Sine Function (SinPoint-SSF) with the initial phase as the origin to deform the point cloud. The other is to use Multiple Sine Function (SinPoint-MSF), with different anchor points as the initial phase. The sine transforms of different parameters are superimposed to obtain richer deformations. Finally, we propose a Markov chain augmentation framework. We further get more semantically consistent samples by random combination of multiple basic transformations, and expand the distribution space of training samples. We experimentally demonstrate that SinPoint outperforms the state-of-the-art point cloud augmentation method on multiple datasets.

Our main contributions can be summarized as follows:

- We analyze the data augmentation from a statistical perspective. This expands the distribution boundary of the dataset and increases its variance.

- We prove that the proposed Sine-based mapping function is a homeomorphism. In theory, it increases the diversity of point clouds without destroying the topological structure.

- We propose a new Markov chain augmentation framework that increases sample diversity by randomly combining different foundational transformations to expand the distribution space of the dataset. expand the dataset's distribution space.

- We demonstrate the effectiveness of our framework by showing consistent improvements over state-of-the-art augmentation methods on both synthetic and real-world datasets in 3D shape classification and part segmentation tasks.

## 2. Related Work

**Deep learning on point cloud.** PointNet (Qi et al., 2017a) is a pioneering work that uses shared MLPs to encode each point individually and aggregates all point features through

global pooling. Inspired by CNNs, PointNet++ (Qi et al., 2017b) adopts a hierarchical multi-scale or weighted feature aggregation scheme to get local features. DGCNN (Wang et al., 2019b) introduces EdgeConv, which utilizes edge features from the dynamically updated graph. Additionally, various works have focused on point-wise multi-layer perceptron (Liu et al., 2020; Xu et al., 2021b; Shen et al., 2018; Ma et al., 2022; Qian et al., 2022; Zhang et al., 2023), convolution (Li et al., 2018; Wu et al., 2019; Thomas et al., 2019; Lin et al., 2020a; Xu et al., 2021a; Liu et al., 2019) , and graph-based (Simonovsky & Komodakis, 2017; Wang et al., 2019a; Lin et al., 2020b; 2021) methods to process point clouds. These methods consistently use Conventional Data Augmentation (CDA) (Qi et al., 2017a;b; Wang et al., 2019b) to improve the model's robust kernel generalization performance, but the improvement is relatively marginal. Parallel to these approaches, other recent works (Chen et al., 2020; Li et al., 2020; Lee et al., 2022; 2021; Zhang et al., 2022; Kim et al., 2021; Ren et al., 2022; Hong et al., 2023) focus on data augmentation to improve the generalization power of deep neural networks in point clouds.

**Data augmentation on point cloud.** Current point cloud augmentation methods can be divided into two categories: self-augmentation and mix-augmentation. Self-augmentation through geometric transformation to augment the shape diversity of the point cloud. For instance, CDA (Qi et al., 2017a;b; Wang et al., 2019b) encompasses geometric transformations like rotation, scaling, translation, and jittering, alongside the addition of noise and point reduction to enhance sample diversity. PointAugment (Li et al., 2020) learn the transformation matrix with an augmented network to produce augmentations. PointWOLF (Kim et al., 2021) selects various anchor points to serve as central points for the local point cloud's weighted transformation, leading to smooth and varied non-rigid deformations. Mix-augmentation uses different strategies to cut and combine two point clouds to form a new point cloud that contains two local shapes. For example, PointMixup (Chen et al., 2020) recently used the shortest path linear interpolation between instances to augment data in the point cloud. PointCutMix (Zhang et al., 2022) benefits from Cut-Mixup and PointMixup, and proposes cutting and pasting of point cloud parts. SageMix (Lee et al., 2022) proposes a saliency-guided Mixup for point clouds to preserve salient local structures.

## 3. Data Augmentation Effectiveness Analysis

Augmented data is obtained through a series of transformations or operations on the original data. Here, we analyze the properties of the augmented data distribution in terms of its variance. First, we define the variance of the given data as follows:

**Definition 1. (Data Variance)** For the training dataset $X = \{x_1, x_2, ..., x_n\}$, the variance of the model's prediction $f(x_i)$ for each sample $x_i$ is expressed as:

$$\text{Var}_X = \frac{1}{n} \sum_{i=1}^{n} \left( \mathbb{E}[f(x_i)] - f(x_i) \right)^2, \tag{1}$$

where, $\mathbb{E}[f(x_i)]$ is the mean of the dataset $X$, $\text{Var}(X)$ is the variance of the model on the dataset $X$, which is a measure of the effect of the volatility of the training data on the model's prediction results.

With data augmentation, the samples in the original dataset are expanded into a variety of different variants. Each augmented sample introduces a slight shift from the original data point, causing the distribution of the data points to spread. These small shifts accumulate with each transformation, increasing the differences between the samples and, consequently, the variance of the dataset. Therefore, we can summarize the theorem as follows:

**Theorem 1. (Data augmentation increases the variance of the dataset)** Given an original dataset $X = \{x_1, x_2, ..., x_n\}$, Suppose that the data augmentation operation $T$ is a random transformation that converts each data point $x_i$ in the original data set into an augmented data point $x'_i$, i.e. $x'_i = T(x_i)$. **(The proof is in the Appendix.)**

$$\text{Var}(X) < \text{Var}(X'). \tag{2}$$

As demonstrated in Theorem 1, generating more diverse data samples extends the coverage of the data distribution, which enhances the model's performance on unseen data through improved accuracy and stability.

## 4. Method

### 4.1. Homeomorphism

The increase in variance reflects the growing diversity of the dataset, which generally helps improve the model's adaptability to different scenarios, thereby reducing overfitting. However, the increase in variance should be moderate; in other words, the dataset's diversity should be enhanced appropriately to boost generalization, but not excessively. However, existing methods do not address the issue of topological consistency before and after deformation, leading to a significant difference between the generated samples and real samples, as shown in Figure 1. This increases the variance of the data excessively. To control this, we introduce homeomorphic mapping to ensure semantic consistency before and after data augmentation, thereby constraining the variance changes within the dataset.

Homeomorphism (Derrick, 1973) is an important mathematical tool used to describe the equivalence relations between

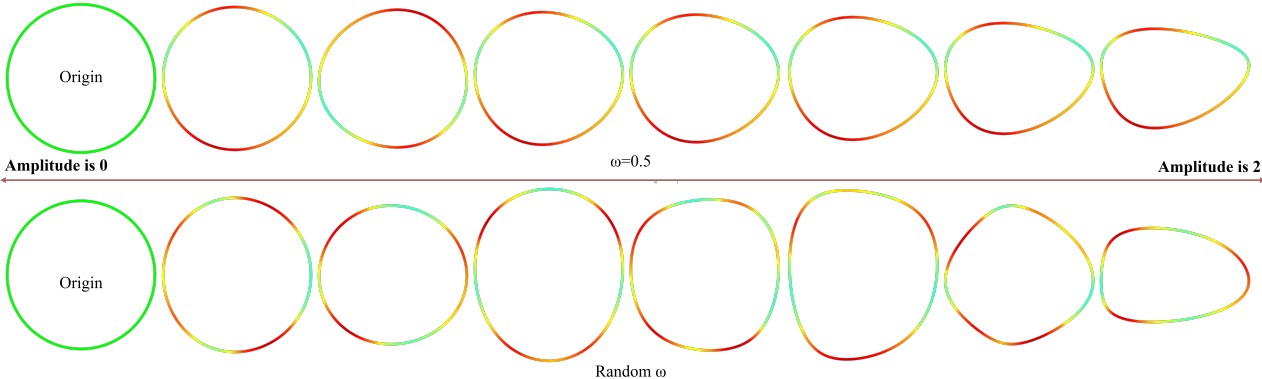

*Figure 3.* SinPoint obtains different degrees of geometric deformation by controlling the amplitude and angular velocity $\omega$. In the above, only two peaks or troughs appear due to the fixed angular velocity $\omega$. In the following, multiple peaks or troughs occur due to the random angular velocity $\omega$. The darker the color, the larger the deformation.

topological spaces. It keeps the proximity of points in space unchanged and makes topological Spaces have the same topological properties.

**Definition 2. (Homeomorphism)** Given two topological spaces $X, Y$, and given a mapping $f : X \rightarrow Y$. $f$ is a homeomorphism of two spaces when it is satisfied that $f$ is a bijection and $f$ and $f^{-1}$ are continuous, denoted as $X \cong Y$.

**Definition 3. (Local homeomorphism)** Let $f : X \rightarrow Y$ is a mapping between two topological spaces $X$ and $Y$. If for every point $x$ in $X$, exists a neighborhood $U$ of $x$ such that $f(U)$ is an open set in $X$ and $f_U : U \rightarrow f(U)$ is a homeomorphism, then $f$ is a local homeomorphism.

They have similar mathematical properties, both of which require the mapping and its inverse mapping to be continuous and maintain the topological structure of the space(Armstrong, 2013).

**Proposition 1. (Topological consistency)** If $f$ is a homeomorphism from $X$ to $Y$, then $X$ and $Y$ have the same topological properties.

**Proposition 2. (Reflexivity, symmetry, and transitivity)** The homeomorphic relation is an equivalence relation, and therefore it has reflexivity (any topological space is a homeomorphism to itself), symmetry (if $X \cong Y$, then $Y \cong X$), and transitivity (if $Y \cong Z$, then $X \cong Z$).

Homeomorphism plays a vital role in ensuring topological consistency. First, a homeomorphism is not only one-to-one (bijective) and continuous, but its inverse is also continuous. This guarantees the preservation of the space topology. Second, when processing point clouds, using a homeomorphism can ensure that the basic shape and structure of the point cloud remain unaltered after augmentation or transformation, thus making the augmented data more consistent with the actual scene or object.

### 4.2. Residual Function

Inspired by the deep residual network (He et al., 2016), we focus on obtaining continuous residual coordinates and generating augmented coordinates by adding offsets to the original coordinates. Specifically, for a given point cloud $P = \{p_1, p_2, ..., p_n\}$, we only need to compute the residual coordinates, represented by $P^{'} - P$, thus the augmentation process becomes:

$$P^{'} = P + g(P). \tag{3}$$

A homeomorphism can ensure that the original space and the deformed space have topological consistency. Therefore, it is very important to select the appropriate residual function, which not only gains diversity but also needs to guarantee that the whole mapping is homeomorphic.

### 4.3. SinPoint

To obtain topologically consistent deformation, we have chosen to use the Sine function as our residual function. The inherent periodic nature of the Sine function allows us to adjust the number of regions that are deformed with precision. Additionally, by manipulating the amplitude of the Sine function, we can precisely control the intensity of the deformation. This displacement field, generated by the Sine function, effectively distorts and deforms specific local regions of the point cloud data without altering the overall topology. As a result, the augmented point cloud data contains more intricate and detailed local features. The standard Sine function is shown below:

$$g(x) = A sin(\omega x + \varphi), \tag{4}$$

where $A$, $\omega$, and $\varphi$ represent the amplitude, the angular velocity (control period), and the initial phase, respectively. The displacement field generated by Sine function is introduced into a homeomorphism, and a homeomorphism based on Sine function can be obtained.

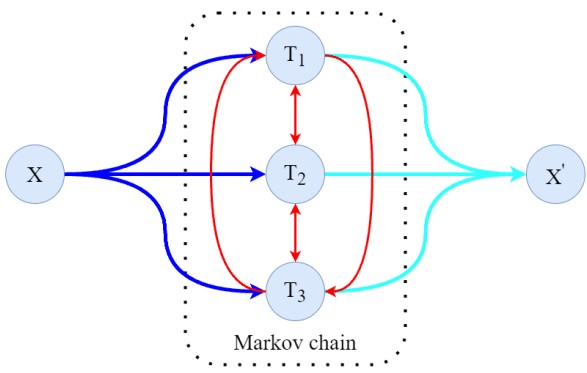

*Figure 4.* Markov Chain Augmentation Process. $X_0$ is the input sample, $X_T$ enhances the sample l, and $T_1 - T_3$ is a different transformation. By randomly transferring input samples through Markov process, more diverse enhanced samples can be obtained.

**Theorem 2. (Homeomorphism Based on Sine Function)**
Given two topological spaces $X, Y$, and given a mapping $f : X \to Y = X + Asin(\omega X + \varphi)$, if $-1 \leq A\omega \leq 1$, then $f$ is a homeomorphism, else $f$ is a local homeomorphism. (**The proof is in the Appendix.**)

Since $f$ is a homeomorphism, we can use it to augment the point cloud and ensure its topological consistency. Meanwhile, this is a class-preserving transformation, ensuring the consistency of the label before and after the deformation. Given a set of points $P = \{p_1, p_2, ..., p_n\}$, where $N$ represents the number of points in the Euclidean space $(x, y, z)$. SinPoint applies a homeomorphism and the resulting augmented point cloud $P'$ is given as follows:

$$P' = P + Asin(\omega P + \varphi), \tag{5}$$

where $Asin(\omega P + \varphi)$ is displacement field of $P$. We need to adjust $A$ and $\omega$ to produce more diverse point clouds. In this paper, we set $A \sim U(-a, a)$ and $\omega \sim U(-w, w)$ to obey the uniform distribution. In this way, more samples with smooth deformation can be generated, which makes the distribution of samples more uniform.

As illustrated in Figure 3, the transformation of the circle by Equation (5) results in continuous local indentations on the point cloud surface, attributable to the Sine function's periodic nature. This deformation technique enables the simulation of a concavity similar to that observed when an object is indented while preserving the point cloud's topology structure.

Since classification tasks are sensitive to global shapes, segmentation tasks rely on local structures. We designed two transformation strategies for SinPoint, SinPoint-SSF based on a single sine function for classification and SinPoint-MSF based on the superposition of multiple sine functions

---

**Algorithm 1** SinPoint Without Markov

**Input:** Original point clouds $P = \{p_1, p_2, ..., p_n\}$
      Condition $key$ for SinPoint-SSF or SinPoint-MSF
      Anchor points number $k$, Amplitude $a$
      Angular velocity $\omega$
**Output:** $P'$
  **if** $key ==$ "SSF"
    Sample $A \sim U(-a, a)$
    Sample $\omega \sim U(-w, w)$
    $P' \leftarrow P + Asin(\omega P)$
  **else if** $key ==$ "MSF"
    **if** using farthest point sampling
      $p_i \leftarrow FPS(P, k)$, #$FPS()$ *is farthest point sampling*
    **else if** using random point sampling
      $p_i \leftarrow RPS(P, k)$, #$RPS()$ *is random point sampling*
    **end if**
    Sample $A_i \sim U(-a, a), i = 1 : k$
    Sample $\omega_i \sim U(-w, w), i = 1 : k$
    $P' \leftarrow P + \frac{1}{k}\sum_{i=1}^{k}\{A_i sin(\omega_i P + p_i)\}$
  **end if**
  **Return** $P'$

---

for segmentation. Algorithm 1 is the process of generating augmented samples.

**SinPoint-SSF** uses a single sine function to perturb the coordinates of the point cloud. We normalize the point cloud to the unit sphere space and take the sphere's center as the initial phase, that is, $\varphi = 0$. Then the SinPoint-SSF transformed point cloud can be expressed as follows:

$$P' = P + Asin(\omega P). \tag{6}$$

**SinPoint-MSF** superposes multiple sine functions to perturb the point cloud. Multiple sinusoidal complex waves exhibit rich waveform characteristics through diverse combinations of frequency, amplitude and phase. SinPoint-MSF first selects $k$ anchor points as the initial phase and samples different amplitudes and angular velocities for additive perturbations. This provides more diversity to the point cloud and generates realistic samples. The transformation of SinPoint-MSF as follows:

$$P' = P + \frac{1}{k}\sum_{i=1}^{k}\{A_i sin(\omega_i P + \varphi_i)\}, \; \varphi_i = p_i. \tag{7}$$

### 4.4. Markov Chain Augmentation Process

Markov process is a random process without memory, that is, the future state depends only on the present state and has nothing to do with the past history. According to Theorem 1, more effective augmented samples are beneficial to

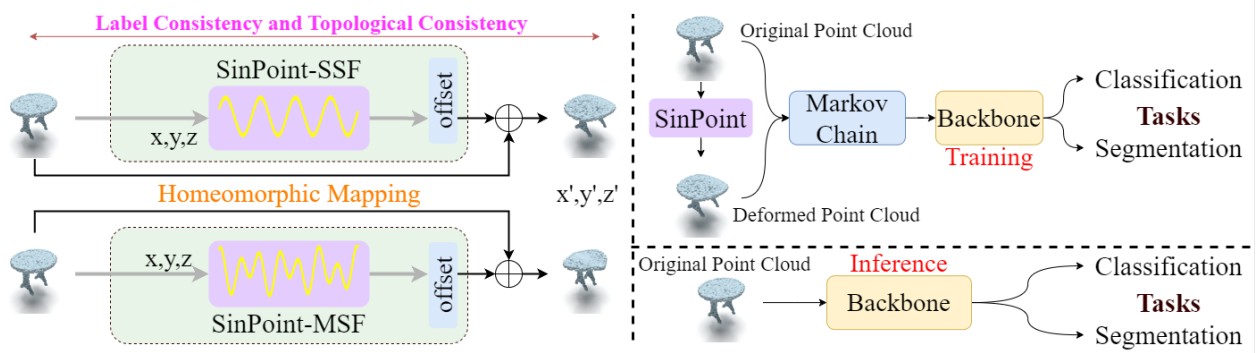

*Figure 5.* An overview of our SinPoint framework. We use SinPoint to get the augmented point cloud and input it into the network with the original point cloud for training. SinPoint can be adapted for a variety of tasks due to label consistency and topological consistency.

the model. To generate more diverse augmented samples and further enhance the model's performance, we propose a Markov Chain Augmentation Framework, as shown in Figure 4, inspired by the general approach of combining various types of transformations. We treat each transformation as a transition matrix, and through multiple transitions, we can obtain a combination of different transformations to further increase the diversity of the data. For example, the transformation sequence may be $\{T_1\}, \{T_1, T_2\}, \{T_1, T_2, T_3\}$.

It is common practice to select a set of augmentations (e.g., for point clouds: rotation, translation, scaling, dropout, etc.) and generate synthetic examples by applying a sequence of these augmentations to an original data point. To model this process, we propose the following procedure: given a data point, we randomly choose augmentations from a set and apply them sequentially. To prevent excessive deviation from the original data, there is a certain probability that we discard the point and restart from a random point in the original dataset. We formalize this approach below.

**Definition 4. (Markov Chain Augmentation Process)** Given a dataset of $n$ examples $X = \{x_1, x_2, ..., x_n\}$, we augment the dataset via augmentation transformations $T_1, T_2, ..., T_m$, which are stochastic transition matrices over a finite state space of possible labeled (augmented) examples. We model this via a discrete time Markov chain with the transitions: The probability of the transition is set to $1/m$, and the maximum number of transitions is $m$.

From Definition 4, by conditioning on the chosen transition, it is clear that the entire process is equivalent to a Markov chain. Note that the transition matrices $T_j$ do not need to be explicitly materialized; instead, they are implicit based on the description of the augmentation.

### 4.5. Framework

Our complete framework is shown in Figure 5. Our framework includes SinPoint, Markov process and sample mixing.

SinPoint maps the point cloud input to a feature space with topological consistency. By incorporating augmented inputs generated by SinPoint, the training process optimizes the backbone's parameters in a larger feature space than the baseline. Markov process uses multiple basic affine transformations to further expand the sample space. **Note that the Markov process and sample mixing are attached to the SinPoint.** During the training stage, by utilizing the same loss function as the baseline and optimizing the model in this new augmented feature space, our method learns a more expressive representation that better fits the input data, leading to improved generalization. During inference, the model uses the original point cloud input to preserve the data's geometric structure, ensuring no disruption to geometric priors in practical applications. Our method maintains point cloud geometric and topological properties. It also extracts more discriminative features, significantly enhancing overall model performance.

## 5. Experiments

In this section, we demonstrate the effectiveness of our proposed method, SinPoint, with various benchmark datasets and baselines. First, for 3D shape classification, we evaluate the generalization performance and robustness using SinPoint-SSF in classification. Next, we compare our SinPoint-MSF with existing data augmentation methods in part segmentation. More ablation studies and implementation details are provided in **Appendix**.

**Datasets.** For classification task, we use two synthetic datasets: ModelNet40 (Wu et al., 2015) and ReducedMN40, and two real-world datasets from ScanObjectNN (Uy et al., 2019): OBJ_ONLY and PB_T50_RS. For the part segmentation task, we adopt a synthetic dataset, ShapeNetPart (Yi et al., 2016).

**Baselines.** For a fair comparison with different data augmentation methods, we use the same backbone network:

*Table 1.* 3D shape classification performance on ModelNet40 /ScanObjectNN using SinPoint-SSF

| Model | Method | Synthetic Datasets | | Real-world Datasets | |
|---|---|---|---|---|---|
| | | ModelNet40 | ReducedMN40 | OBJ_ONLY | PB_T50_RS |
| PointNet | Base | 89.2 | 81.9 | 76.1 | 64.1 |
| | +PointAugment (Li et al., 2020) | 90.9 | 84.1 | 74.4 | 57.0 |
| | +PointMixup (Chen et al., 2020) | 89.9 | 83.4 | - | - |
| | +PatchAugment (Sheshappanavar et al., 2021) | 90.9 | - | - | - |
| | +PointWOLF (Kim et al., 2021) | 91.1 | 85.7 | 78.7 | 67.1 |
| | +RSMix (Lee et al., 2021) | 88.7 | - | - | - |
| | +SageMix (Lee et al., 2022) | 90.3 | - | 79.5 | 66.1 |
| | +PointCutMix (Zhang et al., 2022) | 90.5 | - | - | - |
| | +WOLFMix (Ren et al., 2022) | 90.7 | - | - | - |
| | +PCSalMix (Hong et al., 2023) | 90.5 | - | - | - |
| | +PointPatchMix (Wang et al., 2024) | 90.1 | - | - | - |
| | **+SinPoint(Ours)** | **91.3 (↑ 2.1)** | **86.5 (↑ 4.6)** | **82.6 (↑ 6.5)** | **70.8 (↑ 6.7)** |
| PointNet++ | Base | 90.7 | 85.9 | 84.3 | 79.4 |
| | +PointAugment (Li et al., 2020) | 92.9 | 87.0 | 85.4 | 77.9 |
| | +PointMixup (Chen et al., 2020) | 92.3 | 88.6 | 88.5 | 80.6 |
| | +PatchAugment (Sheshappanavar et al., 2021) | 92.4 | - | 87.1 | 81.0 |
| | +PointWOLF (Kim et al., 2021) | 93.2 | 88.7 | 89.7 | 84.1 |
| | +RSMix (Lee et al., 2021) | 91.6 | - | - | - |
| | +SageMix (Lee et al., 2022) | 93.3 | - | 88.7 | 83.7 |
| | +PointCutMix (Zhang et al., 2022) | 93.4 | - | - | - |
| | +WOLFMix (Ren et al., 2022) | 93.1 | - | - | - |
| | +PCSalMix (Hong et al., 2023) | 93.1 | - | - | - |
| | +PointPatchMix (Wang et al., 2024) | 92.9 | - | - | - |
| | **+SinPoint(Ours)** | **93.4 (↑ 2.7)** | **89.6 (↑ 3.7)** | **90.2 (↑ 5.9)** | **84.5 (↑ 5.1)** |
| DGCNN | Base | 92.2 | 87.5 | 86.2 | 77.3 |
| | +PointAugment (Li et al., 2020) | 93.4 | 88.3 | 83.1 | 76.8 |
| | +PointMixup (Chen et al., 2020) | 92.9 | 89.0 | - | - |
| | +PatchAugment (Sheshappanavar et al., 2021) | 93.1 | - | 86.9 | 79.1 |
| | +PointWOLF (Kim et al., 2021) | 93.2 | 89.3 | 88.8 | 81.6 |
| | +RSMix (Lee et al., 2021) | 93.5 | - | - | - |
| | +SageMix (Lee et al., 2022) | 93.6 | - | 88.0 | 83.6 |
| | +PointCutMix (Zhang et al., 2022) | 93.2 | - | - | - |
| | +WOLFMix (Ren et al., 2022) | 93.2 | - | - | - |
| | +PCSalMix (Hong et al., 2023) | 93.2 | - | - | - |
| | **+SinPoint(Ours)** | **93.7 (↑ 1.5)** | **90.1 (↑ 2.6)** | **90.2 (↑ 4.0)** | **84.6 (↑ 7.3)** |

including PointNet (Qi et al., 2017a), PointNet++ (Qi et al., 2017b), and DGCNN (Wang et al., 2019b). These backbones can more clearly show the impact of data augmentation on model performance. To further verify the validity of SinPoint, we have added a comparison of various backbone networks in the Appendix.

### 5.1. 3D Shape Classification

**Comparisons with SOTA Methods**. Experimental results of 3D shape classification are shown in Table 1. We report the Overall Accuracy (OA) of each model on all four datasets. From the results, we can clearly see that our Sin-Point significantly outperforms all of the previous methods in every dataset and model. Particularly, the average OA

improvement on the synthetic datasets is **2.6%**, and the average OA improvement on the real-world datasets is even **5.9%**, and the maximum improvement was DGCNN reaching **7.3%** in PB_T50_RS. It proves that our SinPoint is more efficient on real data sets. These consistent improvements demonstrate the effectiveness of our framework.

**3D shape classification performance under Various Network Backbones.** The effectiveness of SinPoint is further validated across a variety of network architectures in ModelNet40 (Wu et al., 2015) and ScanObjectNN (Uy et al., 2019), including PointNet (Qi et al., 2017a), PointNet++ (Qi et al., 2017b), DGCNN (Wang et al., 2019b), RSCNN (Liu et al., 2019), PointConv (Wu et al., 2019), PointCNN (Li et al., 2018), GDANet (Xu et al., 2021b), PCT (Guo

*Table 2.* 3D shape classification performance in various architectures on ModelNet40.

| Model | PointNet | PointNet++ | DGCNN | RSCNN | PointConv | PointCNN | GDANet | PCT | PointMLP |
|---|---|---|---|---|---|---|---|---|---|
| Param. | - | 1.4M | 1.8M | - | 18.6M | - | - | 2.8M | 12.6M |
| Base | 89.2 | 90.7 | 92.2 | 91.7 | 92.5 | 92.5 | 93.4 | 93.2 | 94.1 |
| +SinPoint | **91.3 (↑ 2.1)** | **93.4 (↑ 2.7)** | **93.7 (↑ 1.5)** | **92.9 (↑ 1.2)** | **92.8 (↑ 0.3)** | **93.2 (↑ 0.7)** | **93.6 (↑ 0.2)** | **93.5 (↑ 0.3)** | **94.3 (↑ 0.2)** |

et al., 2021) and PointMLP (Ma et al., 2022), PointNeXt-S (Qian et al., 2022), PointMetaBase-S (Lin et al., 2023), SPoTr (Park et al., 2023). From Table 2, we observe that SinPoint has a consistent improvement of accuracy against the baselines (+0.2∼2.7%). Notably, using the basic Point-Net++ and DGCNN, we can surpass the Transformer-based baseline PCT while reducing the parameters by half. This reduction compensates for the parameter deficiency in the network through data augmentation alone. Surprisingly, DGCNN+SinPoint is only 0.4% lower than PointMLP, but the parameters are 7 times lower.

### 5.2. 3D Shape Part Segmentation

Next, we test SinPoint for 3D shape part segmentation task on the ShapeNetPart (Yi et al., 2016) benchmark. We follow the settings from PointNet, PointNet++ and DGCNN that randomly select 2048 points as input for a fair comparison. We compare our methods with several recent methods, including PointMixup (Chen et al., 2020), RSMix (Lee et al., 2021), SageMix (Lee et al., 2022) and PointWOLF (Kim et al., 2021). Table 3 shows that on ShapeNetPart, SinPoint-MSF consistently improves mean IoU (mIoU) over baselines (1.0% over PointNet++ and 0.7% over DGCNN), demonstrating the applicability of SinPoint-MSF to point-wise tasks. Clearly, SinPoint-MSF achieves the SOTA performance in part segmentation. Furthermore, in the **Appendix**, we offer comparisons against a variety baseline models, where our SinPoint consistently outperforms others. Finally, in Figure 6, we present the visualization results for Sinpoint and baseline. Our method can improve the segmentation performance of the model in detail.

### 5.3. Ablation Studies

**Robustness.** Additional studies demonstrate our SinPoint improves the robustness of models against previous methods (Chen et al., 2020; Lee et al., 2021; 2022) on four types of corruption: (1) **Gaussian noise** with ($\sigma \in (0.01, 0.05)$, (2) **Rotation** $180^o$ (X-axis,Z-axis), (3) **Scaling** with a factor in 0.6, 2.0, and (4) **Dropout** with a rate $r \in \{0.25, 0.50\}$. We adopt DGCNN and OBJ_ONLY to evaluate the robustness of models. As shown in Table 4, SinPoint consistently improves robustness in various corruptions. DGCNN with SinPoint shows the best robustness with significant gains compared to previous methods. Importantly, the gain over the baseline significantly increases as the amount of corrup-

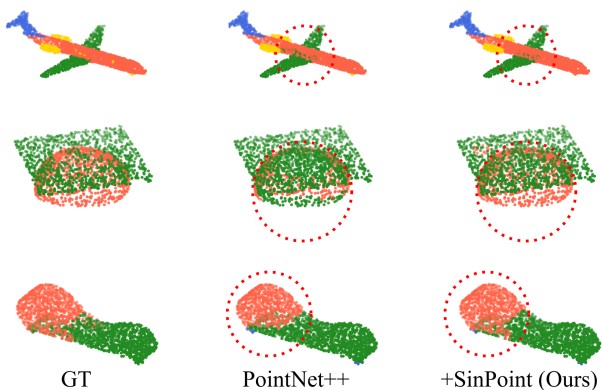

| GT | PointNet++ | +SinPoint (Ours) |

*Figure 6.* 3D part segmentation experiment visualization results.

tion increases: 13.1% for Gaussian noise ($\sigma$ : 0.05), 11.7% for Rotation 180° (Z-axis), 2.6% for Scaling in 2.0, and 3.8% for Dropout (r = 0.5). We believe that the diverse samples augmented by a homeomorphism in SinPoint help models to learn more robust features against various corruptions.

**Ablation study of modules.** Table 5 summarizes the results of the ablation study on DGCNN. Model A gives a baseline classification accuracy of 91.7%. On top of Model A, we use a combination of different augmentations. From the results shown in Table 5, we can see that each augmentation function contributes to producing more effective augmented samples. It is worth noting that when only SinPoint is used, the results already surpass A, and B, while using a mixture of original and augmented samples can again improve the generalization performance of DGCNN. Moreover, when using more modules, the model's generalization ability is further improved to 93.7%, an improvement of 2.0% over the vanilla. This means that the more effective the augmented samples, the greater the improvement of the model, and Theorem 1 is further confirmed experimentally.

**Ablation study of SinPoint with SSF and MSF.** We further verify the performance difference between SinPoint-SSF and SinPoint-MSF in the classification and segmentation tasks. As shown in Table 6, SinPoint-SSF is suitable for classification tasks, and SinPoint-MSF is suitable for segmentation tasks. In classification tasks, a single sine transformation can better enhance the global characteristics of the data, avoid unnecessary complexity, and help improve classification performance. In the segmentation tasks, the

*Table 3.* Complete part segmentation results (mIoU) on ShapeNetPart (Yi et al., 2016) using SinPoint-MSF.

| Model | Base | +CDA | +PointMixup | +RSMix | +SageMix | +PointWOLF | **+SinPoint(Ours)** |
|---|---|---|---|---|---|---|---|
| PointNet++ | 84.8 | 85.1 | 85.5 | 85.4 | 85.7 | 85.2 | **85.8 (↑ 1.0)** |
| DGCNN | 84.8 | 85.0 | 85.3 | 85.2 | 85.4 | 85.2 | **85.5 (↑ 0.7)** |

*Table 4.* Robustness with DGCNN (Wang et al., 2019b) on OBJ_ONLY (Uy et al., 2019)

| Method | Gaussian noise | | Rotation 180° | | Scaling | | Dropout | |
|---|---|---|---|---|---|---|---|---|
| | $\sigma$:0.01 | $\sigma$:0.05 | X-axis | Z-axis | ×0.6 | ×2.0 | 25% | 50% |
| DGCNN | 84.9 | 48.4 | 32.5 | 32.4 | 73.7 | 73.0 | 83.3 | 75.7 |
| + PointMixup | 85.0 | 61.3 | 31.7 | 32.7 | 73.8 | 73.0 | 84.2 | 74.9 |
| + RSMix | 84.2 | 49.1 | 32.7 | 32.6 | 75.0 | 74.5 | 84.0 | 73.6 |
| + SageMix | 85.7 | 51.2 | 36.5 | 37.9 | 75.6 | 75.2 | 84.9 | 79.0 |
| **+ SinPoint** | **85.9** | **61.5** | **38.6** | **44.1** | **76.1** | **75.6** | **85.1** | **79.5** |

*Table 5.* Ablation study of SinPoint on ModelNet40 (Wu et al., 2015). Mix: mixed training samples.

| DGCNN | SinPoint | Markov | Mix | OA | Inc.↑ |
|---|---|---|---|---|---|
| A | | | | 91.7 | - |
| B | | ✓ | | 92.7 | 1.0 |
| C | ✓ | | | **92.9** | **1.2** |
| D | ✓ | ✓ | | **93.0** | **1.3** |
| E | ✓ | | ✓ | **93.4** | **1.7** |
| F | | ✓ | ✓ | **93.2** | **1.5** |
| G | ✓ | ✓ | ✓ | **93.7** | **2.0** |

*Table 6.* Ablation study of SinPoint with SSF and MSF.

| DGCNN +SinPoint | OBJ_ONLY (OA) Classification | ShapeNetPart (mIoU) Segmentation |
|---|---|---|
| SSF | **90.2** | 85.3 |
| MSF | 89.8 | **85.5** |

*Table 7.* Ablation study $A$ and $\omega$ sampling.

| | OA | mAcc |
|---|---|---|
| base | 85.829±0.296 | 83.375±0.395 |
| **Uniform** | **89.759±0.431** | **88.636±0.445** |
| Gaussian | 87.607±0.344 | 85.494±0.409 |

*Table 8.* SinPoint on S3DIS and SemanticKITTI.

| Method | S3DIS | SemanticKITTI |
|---|---|---|
| MinkNet | 64.8 | 55.9 |
| **+SinPoint (Ours)** | **65.4 (↑ 0.6)** | **63.5 (↑ 7.6)** |

et al., 2019) performance. This also shows that our SinPoint is also suitable for scene tasks.

# 6. Conclusion

We propose SinPoint, a novel point cloud augmentation framework that combines Sine transformation grounded in homeomorphism and Markov process. The homeomorphism ensures topological consistency between the data before and after the sine transformation, while the Markov process generates more diverse augmented samples through the superposition of multiple transformations. We conducted extensive experiments and demonstrated how SinPoint improves the performance of three representative networks on multiple datasets. Our findings show that the augmentations we produce are visually realistic and beneficial to the models, further validating the importance of our approach to understanding the local structure of point clouds.

# Impact Statement

This paper presents work whose goal is to advance the field of Machine Learning. There are many potential societal consequences of our work, none of which we feel must be specifically highlighted here.

# Acknowledgements

This work was supported by the National Science Fund of China under Grant Nos. 62361166670, U24A20330 and 62276135.

composite transformation of multiple sine transformation superpositions provides more diverse local details, thereby improving the segmentation accuracy.

**Ablation study of amplitude $A$ and angular velocity $\omega$ sampling.** We explore the effectiveness of amplitude $A$ and angular velocity $\omega$ sampling. Table 7 shows the results with various sampling methods for amplitude $A$ and angular velocity $\omega$. Uniform and Gaussian sampling introduce +3.9% and 2.8% gains over base DGCNN. The OA with Uniform sampling is 2.1% higher than Gaussian sampling, which means that uniform sampling leads to greater diversity and maximizes model performance.

**Performance on scene segmentation.** We added additional experiments to the S3DIS (Armeni et al., 2016) and SemanticKITTI (Behley et al., 2019) datasets. As shown in Table 8, our SinPoint is still able to improve MinkNet (Choy

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

## A. Datasets

ModelNet40 is a widely used synthetic benchmark dataset containing 9840 CAD models in the training set and 2468 CAD models in the validation set, with a total 40 classes of common object categories. ReducedMN40 comes from PointMixup (Chen et al., 2020) and only contains 20% training samples to simulate data scarcity. ScanObjectNN is a real-world dataset that is split into 80% for training and 20% for evaluation. Among the variants of ScanObjectNN, we adopt the simplest version (OBJ_ONLY) and the most challenging version (PB_T50_RS). OBJ_ONLY, which has 2,309 and 581 scanned objects for the training and validation sets, respectively, and PB_T50_RS, which is a perturbed version with 11,416 and 2,882 scanned objects for the training and validation sets, respectively. Both have 15 classes. We use only coordinates $(x, y, z)$ of 1024 points for training models without additional information, such as the normal vector. For the part segmentation task, we adopt a synthetic dataset, **ShapeNetPart** (Yi et al., 2016), which contains 14,007 and 2,874 samples for training and validation sets. ShapeNetPart consists of 16 classes with 50 part labels. Each class has 2 to 6 parts.

## B. A Implementation Detail

We conduct experiments using Python and PyTorch with two NVIDIA TITAN RTX for point clouds. Following the original configuration in (Qi et al., 2017a;b; Wang et al., 2019b), we use the SGD optimizer with an initial learning rate of $10^{-1}$ and weight decay of $10^{-3}$ for PointNet (Qi et al., 2017a) and PointNet++ (Qi et al., 2017b) and SGD with an initial learning rate of $10^{-2}$ and weight decay of $10^{-4}$ for DGCNN (Wang et al., 2019b). We train models with a batch size of 32 for 300 epochs. For hyperparameters of SinPoint-SSF and SinPoint-MSF, we opt $A = 0.6, w = 2.5, k = 4$ in the entire experiment. In the Markov chain augmentation process, we choose scaling, shifting, rotation and jittering as the base transformation, then $m = 4$ and the transition probability is 25%.

## C. Proof

**Theorem 1. (Data augmentation increases the variance of the dataset)** Given a original dataset $X = \{x_1, x_2, ..., x_n\}$, Suppose that the data augmentation operation $T$ is a random transformation that converts each data point $x_i$ in the original data set into an augmented data point $x_i'$, i.e. $x_i' = T(x_i)$. So $\text{Var}(X) < \text{Var}(X')$.

**Proof:** Suppose that the transformation $T$ introduces a random disturbance to each data point $\epsilon_i$, i.e. :

$$x_i' = x_i + \epsilon_i, \tag{8}$$

where $\epsilon_i$ is a random variable from a distribution, representing random changes in the data augmentation process. For computational simplicity, we assume that the mean of this random disturbance is zero, that is, $E[\epsilon_i] = 0$, and the variance is $\text{Var}(\epsilon_i) = \rho_{\epsilon_i}^2$.

Next, we calculate the variance $\text{Var}(X')$ of the augmented dataset:

$$\text{Var}(X') = \frac{1}{n} \sum_{i=1}^{n} (x_i' - \mu_X')^2 = \frac{1}{n} \sum_{i=1}^{n} (x_i + \epsilon_i - \mu_X)^2 \tag{9}$$

where $E[X] = \mu_X$. Expanded square term:

$$\text{Var}(X') = \frac{1}{n} \sum_{i=1}^{n} \left[ (x_i - \mu_X)^2 + 2(x_i - \mu_X)\epsilon_i + \epsilon_i^2 \right] \tag{10}$$

We break this formula into three categories:

$$\text{Var}(X') = \frac{1}{n} \sum_{i=1}^{n} (x_i - \mu_X)^2 + \frac{1}{n} \sum_{i=1}^{n} 2(x_i - \mu_X)\epsilon_i + \frac{1}{n} \sum_{i=1}^{n} \epsilon_i^2 \tag{11}$$

Notice that $E[\epsilon_i] = 0$, and $\epsilon_i$ is independent of $x_i$, so the second term is zero:

$$\frac{1}{n} \sum_{i=1}^{n} 2(x_i - \mu_X)\epsilon_i = 0 \tag{12}$$

Therefore, the variance of the augmented dataset is simplified to:

$$\text{Var}(X') = \text{Var}(X) + \frac{1}{n}\sum_{i=1}^{n}\epsilon_i^2 \tag{13}$$

Since $\epsilon_i$ is a random disturbance with a variance of $\rho_{\epsilon_i}^2$, therefore:

$$\frac{1}{n}\sum_{i=1}^{n}\epsilon_i^2 \approx \sigma_\epsilon^2 \tag{14}$$

Thus, the variance of the enhanced data set can be expressed as:

$$\text{Var}(X') = \text{Var}(X) + \sigma_\epsilon^2 \tag{15}$$

Since $\rho_{\epsilon_i}^2 > 0$, data augmentation necessarily leads to an increase in variance.

**Theorem 2. (Homeomorphism Based on Sine Function)** Given two topological spaces $X, Y$, and given a mapping $f : X \rightarrow Y = X + Asin(\omega X + \varphi)$, if $-1 \leq A\omega \leq 1$, then $f$ is a homeomorphism, else $f$ is a local homeomorphism.

**1) Proof: if $-1 \leq A\omega \leq 1$, then $f$ is a homeomorphism**

Step 1: *Continuous*

Let $g(x) = x$, $h(x) = Asin(\omega x + \varphi), x \in R$. Since $g(x) = x$ is a continuous function and $h(x) = Asin(\omega x + \varphi)$ is also a continuous function, then $f(x) = g(x) + h(x) = x + Asin(\omega x + \varphi)$ must be a continuous function.

Step 2: *Bijective*

Let $f(x) = x + Asin(\omega x + \varphi), x \in R$. Then $f'(x) = 1 + A\omega cos(\omega x + \varphi), x \in R$.

Since $A\omega cos(\omega x + \varphi) \in [-A\omega, A\omega]$. Next $f'(x) \in [1 - A\omega, 1 + A\omega]$.

Let $f'(x) > 0 => 1 + A\omega cos(\omega x + \varphi) > 0 => 1 - A\omega > 0 => A\omega < 1$.

Let $f'(x) < 0 => 1 + A\omega cos(\omega x + \varphi) < 0 => 1 + A\omega < 0 => A\omega > -1$.

As $-1 \leq A\omega \leq 1$, $f$ is a monotone function. In this case, $\forall A\omega \in [-1, 1], \exists f^{-1}$ is $f$

In this case, $\forall A\omega \in [-1, 1]$, $f$ must be invertible, and the inverse function of $f$ is $f^{-1}$.

Thus, $f$ is bijective if and only if $-1 \leq A\omega \leq 1$.

Finally, when $-1 \leq A\omega \leq 1$, then $f : X \rightarrow Y = X + Asin(\omega X + \varphi)$ is a homeomorphism.

**2) Proof: if $A\omega \in R$, then $f$ is a local homeomorphism**

as we konw, $h(x) = Asin(\omega x + \varphi), x \in R$ is a periodic function. where $T = \frac{2k\pi}{\omega}$.

Let $2k\pi - \frac{pi}{2} \leq \omega x + \varphi \leq 2k\pi + \frac{pi}{2}, k \in Z$, then $\frac{2k\pi - \frac{pi}{2} - \varphi}{\omega} \leq x \leq \frac{2k\pi + \frac{pi}{2} - \varphi}{\omega}$, now $f$ is strictly increasing.

Let $2k\pi + \frac{pi}{2} \leq \omega x + \varphi \leq 2k\pi + \frac{3pi}{2}, k \in Z$, then $\frac{2k\pi + \frac{pi}{2} - \varphi}{\omega} \leq x \leq \frac{2k\pi + \frac{3pi}{2} - \varphi}{\omega}$, now $f$ is strictly decreasing.

Thus, when $A\omega \in R$, $\forall u \in R, \exists U$ such that $f_U$ is monotone and $f$ is a local homeomorphism.

# D. Analysis

## D.1. Detailed Analysis of How Our Method Affects Part Boundaries

The Jacobian determinant is a quantity that describes the 'stretching' or 'shrinking' properties of the mapping locally. In practical applications, when the sign of the Jacobian determinant changes, a transition from 'stretching' to 'shrinking' may occur, leading to a folding phenomenon.

Given the mapping $P' = P + Asin(\omega P + \phi)$, we can see it as a transformation from $P$ to $P'$, where $P = (x, y, z)$ and $P' = (x', y', z')$ denote points in 3D space.

**Calculate the Jacobian.**

Set:
$$g(P) = Asin(\omega P + \phi). \tag{16}$$

So the transformation can be written as follows:
$$P' = P + g(P). \tag{17}$$

To calculate the Jacobian $J_{P'}(P)$ of this transformation, we need to take the derivatives with respect to the components of $P'$ with respect to $P$ separately.

1 Write the component forms of $P'$ Suppose each component form is as follows:
$$x' = x + A_x sin(\omega_x x + \phi_x). \tag{18}$$
$$y' = y + A_y sin(\omega_y y + \phi_y). \tag{19}$$
$$z' = z + A_z sin(\omega_z z + \phi_z). \tag{20}$$

Thus, $P' = (x', y', z')$ is obtained by adding $P = (x, y, z)$ to the sine transform of the components.

2 Calculate the elements of the Jacobian matrix.

The Jacobian matrix $J_{P'}(P)$ is of the form:

$$J_{P'}(P) = \begin{bmatrix} \frac{\partial_{x'}}{\partial_x} & \frac{\partial_{x'}}{\partial_y} & \frac{\partial_{x'}}{\partial_z} \\ \frac{\partial_{y'}}{\partial_x} & \frac{\partial_{y'}}{\partial_y} & \frac{\partial_{y'}}{\partial_z} \\ \frac{\partial_{z'}}{\partial_x} & \frac{\partial_{z'}}{\partial_y} & \frac{\partial_{z'}}{\partial_z} \end{bmatrix} \tag{21}$$

Since $x'$ depends only on $x$, $y'$ depends only on $y$, and $z'$ depends only on $z$, the partial derivative matrix is a diagonal matrix.

Diagonal elements are computed:
$$\frac{\partial_{x'}}{\partial_x} = 1 + A_x \omega_x cos(\omega_x x + \phi_x) \tag{22}$$

$$\frac{\partial_{y'}}{\partial_y} = 1 + A_y \omega_y cos(\omega_y y + \phi_y) \tag{23}$$

$$\frac{\partial_{x'}}{\partial_z} = 1 + A_z \omega_z cos(\omega_z z + \phi_z) \tag{24}$$

So the Jacobian matrix is as follows:
$$J_{P'}(P) = \begin{bmatrix} 1 + A_x \omega_x cos(\omega_x x + \phi_x) & 0 & 0 \\ 0 & 1 + A_y \omega_y cos(\omega_y y + \phi_y) & 0 \\ 0 & 0 & 1 + A_z \omega_z cos(\omega_z z + \phi_z) \end{bmatrix} \tag{25}$$

The determinant of this Jacobian is:
$$det(J_{P'}(P)) = (1 + A_x \omega_x cos(\omega_x x + \phi_x)) \cdot (1 + A_y \omega_y cos(\omega_y y + \phi_y)) \cdot (1 + A_z \omega_z cos(\omega_z z + \phi_z)). \tag{26}$$

If $det(J_{P'}(P)) > 0$, the map is orientation-preserving near that point, that is, no direction flip occurs locally.

If $det(J_{P'}(P)) < 0$, the map is orientation-reversing near that point, that is, the local direction has been flipped.

Thus, when $|Aw| < 1$, $det(J_{P'}(P)) > 0$, not affecting the part boundaries. When $|Aw| > 1$, that is, $|A|$ and $|w|$ are large and in the same direction, the determinant may change sign in different regions, which means that the mapping may alternately hold or flip directions in different regions. At this time, the determinant of some regions approaches zero or becomes negative, which may cause the points in local regions to be compressed or collapsed, affecting the part boundaries. In addition, the degree of specific influence on the boundary is also affected by the parameter. When the parameter value does not change much, this influence can be ignored.

As shown in Figure 7, selecting too large a parameter can result in folding, which may affect **part boundaries**. Therefore, in order to ensure topological consistency and no drastic folding occurs, we choose an appropriate $A$ and $w$.

### D.2. Detailed Analysis of Label Consistency

1) One-to-one mapping between point clouds and labels.

Given point cloud $P = (p_1, p_2, ..., p_n)$. Each point $p_i$ has a corresponding label $l_i$. So you can get a one-to-one correspondence:

$$p_1 \rightarrow l_1, p_2 \rightarrow l_2, ..., p_n \rightarrow l_n. \tag{27}$$

This means that for each point $p_i$, it has a corresponding label $l_i$, which is associated with the geometric position of the point.

2) Definition of homeomorphic mapping.

A homeomorphic map $f(x)$ is a map that preserves topology and changes the positions of points but not the relative relations between them.

If we deform the point cloud by the homeomorphic mapping $f(P)$, then the new point cloud $P'$ is:

$$P' = P + f(P). \tag{28}$$

Therefore, each point position of the new point cloud $P'$ is $p_i' = p_i + f(p_i)$, that is, each point $p_i$ moves to the new position $p_i'$ by mapping $f(x)$.

3) Label consistency.

Since the homeomorphic mapping one-to-one correspondence of points in the point cloud, the index and label of each point are kept consistent after the point cloud is deformed. In other words, the label $l_i$ corresponding to the deformed position $p_i'$ does not change. We can obtain the following relationship:

$$p_1' \rightarrow p_1 \rightarrow l_1, p_2' \rightarrow p_2 \rightarrow l_2, ..., p_n' \rightarrow p_n \rightarrow l_n. \tag{29}$$

Thus:

$$p_1' \rightarrow l_1, p_2' \rightarrow l_2, ..., p_n' \rightarrow l_n. \tag{30}$$

This means that in the deformed point cloud $P'$, the label of each point $p_i'$ remains the same as the label of $p_i$ in the original point cloud $P$.

## E. Additional Experiments

### E.1. Mean and Standard Deviation

Performance oscillation is an essential issue in point cloud benchmarks. However, for a fair comparison with the numbers reported in PointMixup (Chen et al., 2020) RSMix (Lee et al., 2021), and SageMix (Lee et al., 2022), we followed the prevalent evaluation metric in point clouds, which reports the best validation accuracy. Apart from this, to make the experiment fair, like SageMix, we provide the additional results with five runs on OBJ_ONLY (Uy et al., 2019) and report the mean and variance of our method, and the experimental results are shown in Table 9.

*Table 9.* Mean and standard deviation measures on OBJ_ONLY

| Method | Model | | |
|---|---|---|---|
| | PointNet | PointNet++ | DGCNN |
| Base | 78.56±0.51 | 86.14±0.39 | 85.72±0.44 |
| +PointMixup | 78.88±0.28 | 87.50±0.26 | 86.26±0.34 |
| +RSMix | 77.60±0.56 | 87.30±0.65 | 85.88±0.59 |
| +SageMix | 79.14±0.30 | 88.42±0.26 | 87.32±0.53 |
| **+SinPoint(Ours)** | **82.21±0.36** | **89.83±0.35** | **88.64±0.55** |

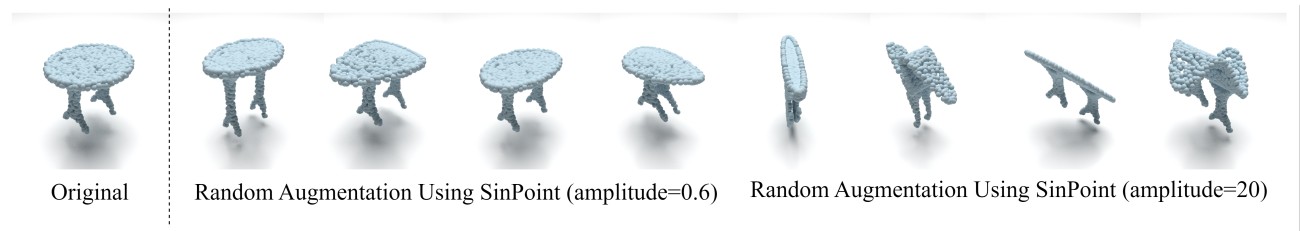

Original    Random Augmentation Using SinPoint (amplitude=0.6)    Random Augmentation Using SinPoint (amplitude=20)

*Figure 7.* Visualization of ablation results for different parameters of SinPoint.

### E.2. Ablation Studies and Analyses

**3D shape classification performance under Various Backbones.** The effectiveness of SinPoint is further validated across some of the latest backbones in ScanObjectNN (Uy et al., 2019), including PointMLP (Ma et al., 2022), PointNeXt-S (Qian et al., 2022), PointMetaBase-S (Lin et al., 2023), SPoTr (Park et al., 2023). As shown in Table 10, our SinPoint can improve the model's performance.

*Table 10.* 3D shape classification performance in various architectures on PB_T50_RS

| Model | PointMLP | PointNeXt-S | PointMetaBase-S | SPoTr |
|---|---|---|---|---|
| Params. (M) | 13.2 | 1.4 | 1.4 | 1.7 |
| FLOPs (G) | 31.3 | 1.6 | 0.6 | 10.8 |
| Throughput (ins./sec.) | 191 | 2040 | 2674 | 281 |
| Base | 85.7 | 87.7 | 88.2 | 88.6 |
| +SinPoint(Ours) | **87.5(↑1.8)** | **88.9(↑1.2)** | **89.3(↑1.1)** | **89.5(↑0.9)** |

**Ablation study of amplitude $A$.** The amplitude $A$ of the Sine function controls the degree of deformation in SinPoint. As shown in Table 11, the larger the $A$, the larger the deformation. However, in Figure 7, too large deformation will lead to the loss of local geometric information. Therefore, we need proper deformation. As demonstrated in Theorem 1, data augmentation amplifies the variance of the data distribution. However, extremes in variance (either excessive or insufficient) degrade model's performance. Consequently, controlling the augmentation intensity within an optimal range becomes imperative.

*Table 11.* Ablation study of amplitude $A$.

| $A$ | 0.2 | 0.4 | 0.6 | 0.8 | 1.0 |
|---|---|---|---|---|---|
| OA (%) | 88.985 | 88.812 | **90.189** | 89.329 | 88.985 |
| mAcc (%) | 87.811 | 87.663 | **89.045** | 88.642 | 88.303 |

**3D part segmentation performance under Various Baselines.** The effectiveness of SinPoint is further validated across a variety of network architectures in ShapeNetPart (Yi et al., 2016), including PointNet (Qi et al., 2017a), PointNet++ (Qi et al., 2017b), DGCNN (Wang et al., 2019b), CurveNet (Xiang et al., 2021), 3DGCN (Lin et al., 2021), GDANet (Xu et al., 2021b), PointMLP (Ma et al., 2022), SPoTr (Park et al., 2023), PointMetaBase (Lin et al., 2023) and DeLA (Chen et al., 2023). Table 12 shows that SinPoint has a consistent improvement of mean Inter-over-Union (mIoU) over the baselines (+0.1∼1.0%).

### E.3. Visualization

**Convergence analysis.** As shown in Figure 8, our SinPoint demonstrates faster convergence during the training phase and achieves higher accuracy than the baseline. Consistent performance improvement is achieved under various parameter Settings.

**Training efficiency.** As shown in Table 13, Our SinPoint achieves the best performance while reducing the time per training

*Table 12.* Part segmentation performance in various architectures on ShapeNetPart (Yi et al., 2016). The proposed SinPoint shows consistent improvements over baselines.

| Model | PointNet | PointNet++ | DGCNN | CurveNet | 3DGCN | GDANet | PointMLP | SPoTr | PointMetaBase | DeLA |
|---|---|---|---|---|---|---|---|---|---|---|
| Base | 83.5 | 84.8 | 84.8 | 86.6 | 86.4 | 86.1 | 85.8 | 87.0 | 86.9 | 87.0 |
| +SinPoint | **84.4 (↑ 0.9)** | **85.8 (↑ 1.0)** | **85.5 (↑ 0.7)** | **86.8 (↑ 0.2)** | **86.6 (↑ 0.2)** | **86.2 (↑ 0.1)** | **86.1 (↑ 0.3)** | **87.2 (↑ 0.2)** | **87.3 (↑ 0.4)** | **87.4 (↑ 0.4)** |

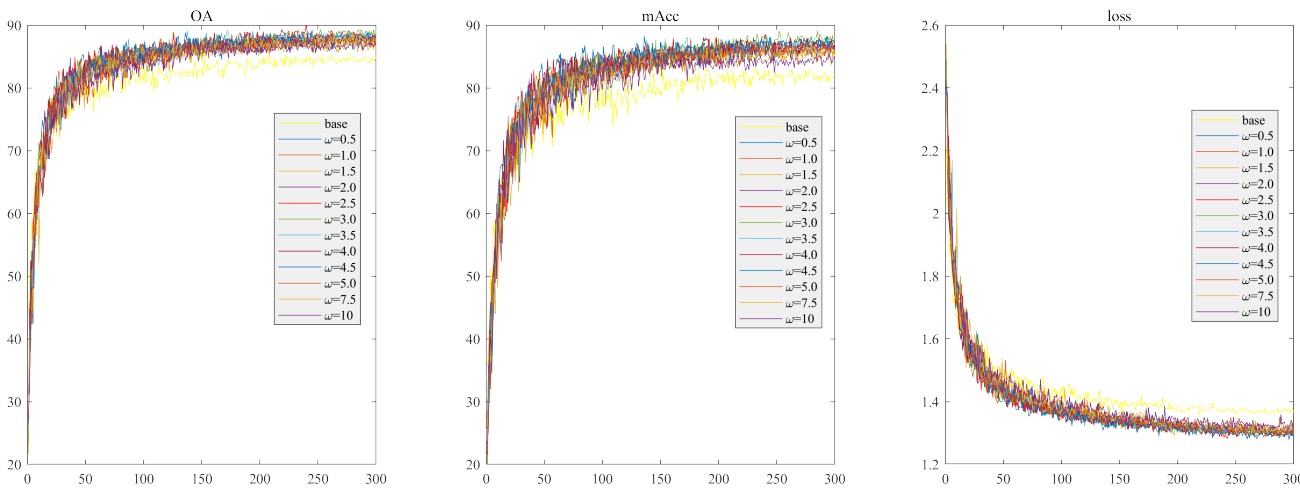

*Figure 8.* Convergence curve during the model training phase. Our SinPoint has a faster convergence speed and higher convergence accuracy than the baseline.

epoch. Notably, the training time is reduced by 10 times compared to PointAugment, 6 times compared to SageMix, and 2 times compared to PointWOLF.

*Table 13.* Comparisons of the training efficiency on ModelNet40 using PointNet.

| Method | PointAugment | SageMix | PointWOLF | SinPoint (Ours) |
|---|---|---|---|---|
| OA (%) | 74.4 | 79.5 | 78.7 | **82.6** |
| Time (sec) | 84 | 51 | 15 | **8** |

**Qualitative results of SinPoint.** In Figure 9, we give a visualization of more augmented samples. In Figure 10, we present an augmented sample visualization comparison between SinPoint-SSF and SinPoint-MSF.

**Qualitative results compare SinPoint with PointWOLF.** We compare our SinPoint with PointWOLF in geometric diversity and topological consistency of point clouds. As can be seen from Figure 11, our SinPoint is entirely superior to PointWOLF and does not require **AugTune** (Kim et al., 2021). The results generated by our SinPoint are more in line with the real world. On the contrary, many of the results generated by PointWOLF are out of the reality.

### E.4. Discussion and Future Work

In the future, we will apply SinPoint to more tasks, such as feature space augmentation, few-shot learning (Liu et al., 2019; Qi et al., 2017a), semantic segmentation (Chen et al., 2019; Xu et al., 2021c; Wang et al., 2019a), object detection (Taha et al., 2020; Zhao et al., 2021; Sugimura et al., 2020), point cloud registration (Wu et al., 2024), etc. It is worth noting, however, that different tasks require different considerations. For example, in few-shot learning, SinPoint maximizes the diversity of training data when samples are extremely scarce. For object detection, SinPoint can generate richer 3D transformations for various object instances in a 3D scene, and so on. Therefore, SinPoint will be easily extended to other tasks.

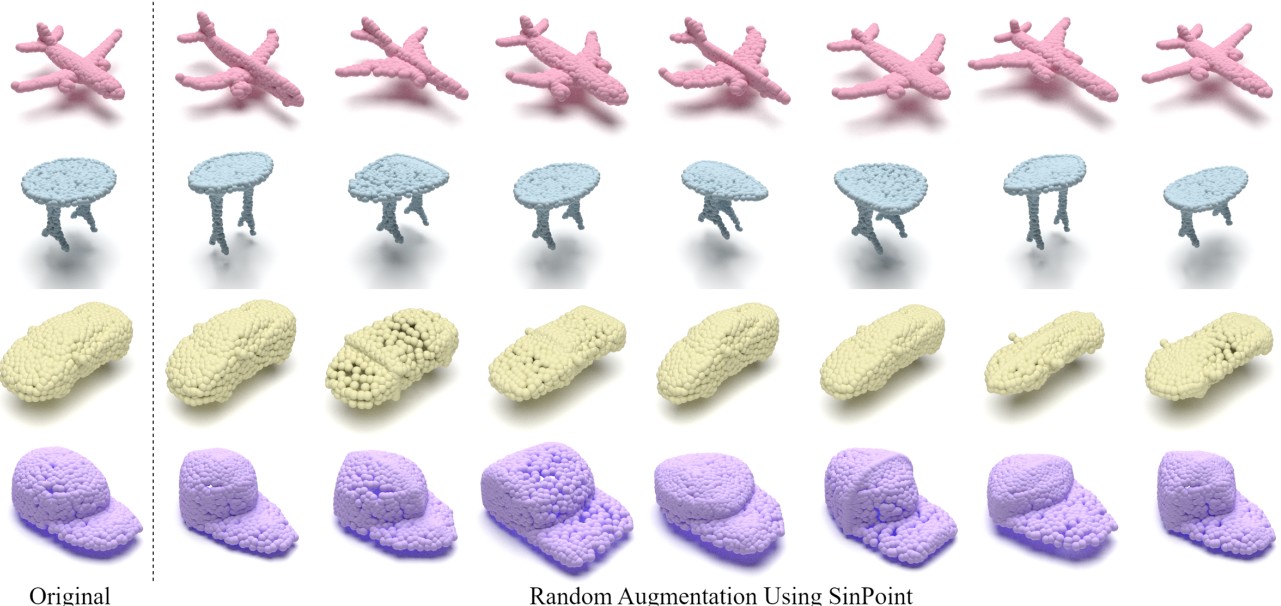

Original                                Random Augmentation Using SinPoint

*Figure 9.* Augmented point clouds using SinPoint. In each row, the left-most sample is the original, and the remaining samples are its transformed results.

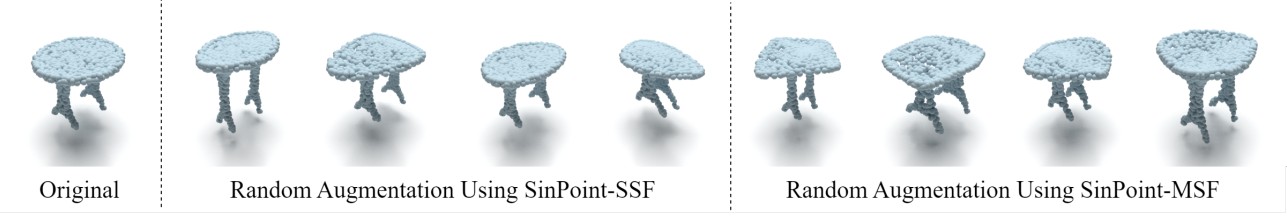

Original        Random Augmentation Using SinPoint-SSF        Random Augmentation Using SinPoint-MSF

*Figure 10.* Visualization comparison of our SinPoint. The left is SinPoint-SSF, right is SinPoint-MSF. SinPoint-MSF can produce varying degrees of local deformation in different regions.

## F. Compare to Other Methods

### F.1. Conventional Data Augmentation

A Conventional Data Augmentation (CDA) (Qi et al., 2017a;b; Wang et al., 2019b) for point clouds applies a global similarity transformation (e.g., scaling, rotation, and translation) and point-wise jittering. Given a set of points $P = \{p_i | i = 1, 2, ..., N\}$, where $N$ represents the number of points in the Euclidean space $(x, y, z)$. The augmented point cloud $P'$ is given as follows:

$$P' = SRP + B. \tag{31}$$

where $S > 0$ is a scaling factor, $R$ is a 3D rotation matrix, and $B \in R^{N \times 3}$ is a translation matrix with global translation and point-wise jittering. Typically, $R$ is an extrinsic rotation parameterized by a uniformly drawn Euler angle for the up-axis orientation. Scaling and translation factors are uniformly drawn from an interval, and point-wise jittering vectors are sampled from a truncated Gaussian distribution.

Obviously, when $B$ does not exist, CDA is a rigid transformation, and when $B$ exists, CDA is simply a similarity transformation with jitter. Thus, CDA cannot simulate diverse shapes and deformable objects, and the enhanced sample has poor diversity.

## F.2. Mix-Augmentation

Several works (Chen et al., 2020; Lee et al., 2021) tried to leverage the Mixup in point cloud. PointMixup linearly interpolates two point clouds by

$$P^{'} = \{\lambda p_i^{\alpha} + (1-\lambda)p_{\phi^*(i)}^{\beta}\}_i^n, y^{'} = \lambda y^{\alpha} + (1-\lambda)y^{\beta}. \tag{32}$$

$$\phi^* = \arg\min_{\phi \in \Phi} \sum_{i=1}^n \|p_i^{\alpha} + p_{\phi(i)}^{\beta}\|_2 \tag{33}$$

where $P^t = \{p_1^t, ..., p_n^t\}$ is the set of points with $t \in \{\alpha, \beta\}$, $n$ is the number of points, and $\phi^* : \{1, ..., n\} \to \{1, ..., n\}$ is the optimal bijective assignment between two point clouds. In RSMix (Lee et al., 2021), they generate an augmented sample by merging the subsets of two objects, defined as $P^{'} = (P^{\alpha} - S^{\alpha}) \cup S^{\beta \to \alpha}$, where $S^t \subset P^t$ $t$ is the rigid subset and $S^{\beta \to \alpha}$ denotes $S^{\beta}$ translated to the center of $S^{\alpha}$. SageMix sequentially selects the query point based on the saliency scores to improve the above method.

Although these methods have shown that Mixup is effective for point clouds, some limitations have remained unresolved: loss of original structures, discontinuity at the boundary, and loss of discriminative regions. Therefore, although the method based on mixup can increase the diversity of samples by mixup different samples, it destroys the point cloud structure.

## F.3. Self-Augmentation

A representative method is PointWOLF. PointWOLF generates deformation for point clouds by a convex combination of multiple transformations with smoothly varying weights. PointWOLF first selects several anchor points and locates random local transformations (e.g., similarity transformations) at the anchor points. Based on the distance from a point in the input to the anchor points, PointWOLF differentially applies the local transformations. The smoothly varying weights based on the distance to the anchor points allow spatially continuous augmentation and generate realistic samples. Given an anchor point $p_j^A \in P^A$. the local transformation for an input point $p_j^A \in P_i$ can be written as:

$$p_i^j = S_j R_j (p_i - p_j^A) + B_j + p_j^A. \tag{34}$$

where $R_j$, $S_j$ and $B_j$ are rotation matrix, scaling matrix and translation vector $bj$ respectively which specifically correspond to $p_j^A$. $S$ is a diagonal matrix with three positive real values, i.e., $S = diag(s_x, s_y, s_z)$ to allow different scaling factors for different axes.

Due to the local rotation and translation, the local separation from the main body will cause the topological structure of the point cloud to change, so PointWOLF is not a homeomorphism and cannot guarantee the topological consistency.

Meanwhile, AugTune is also required due to the poor performance of the PointWOLF direct transform. For $N$ points and $M$ anchor points, the time complexity of PointWOLF is $O(MN) + O(N)$. However, our SinPoint can produce realistic augmented data without interpolation, and our SinPoint time complexity is only $O(N)$ or $O(MN)$, which can reduce the amount of computation.

## F.4. Limitations

Since SinPoint uses homeomorphism computing to augment the point cloud, additional computational effort is required, which is consistent with the framework limitations of all data augmentation. Although we demonstrated that our method can be applied to point cloud classification and part segmentation, other tasks have not yet been investigated using our method, such as scene segmentation and object detection of point cloud datasets of indoor and outdoor. However, because our framework has topological consistency and label consistency, we believe our framework can be extended to different tasks. These are left for future work.

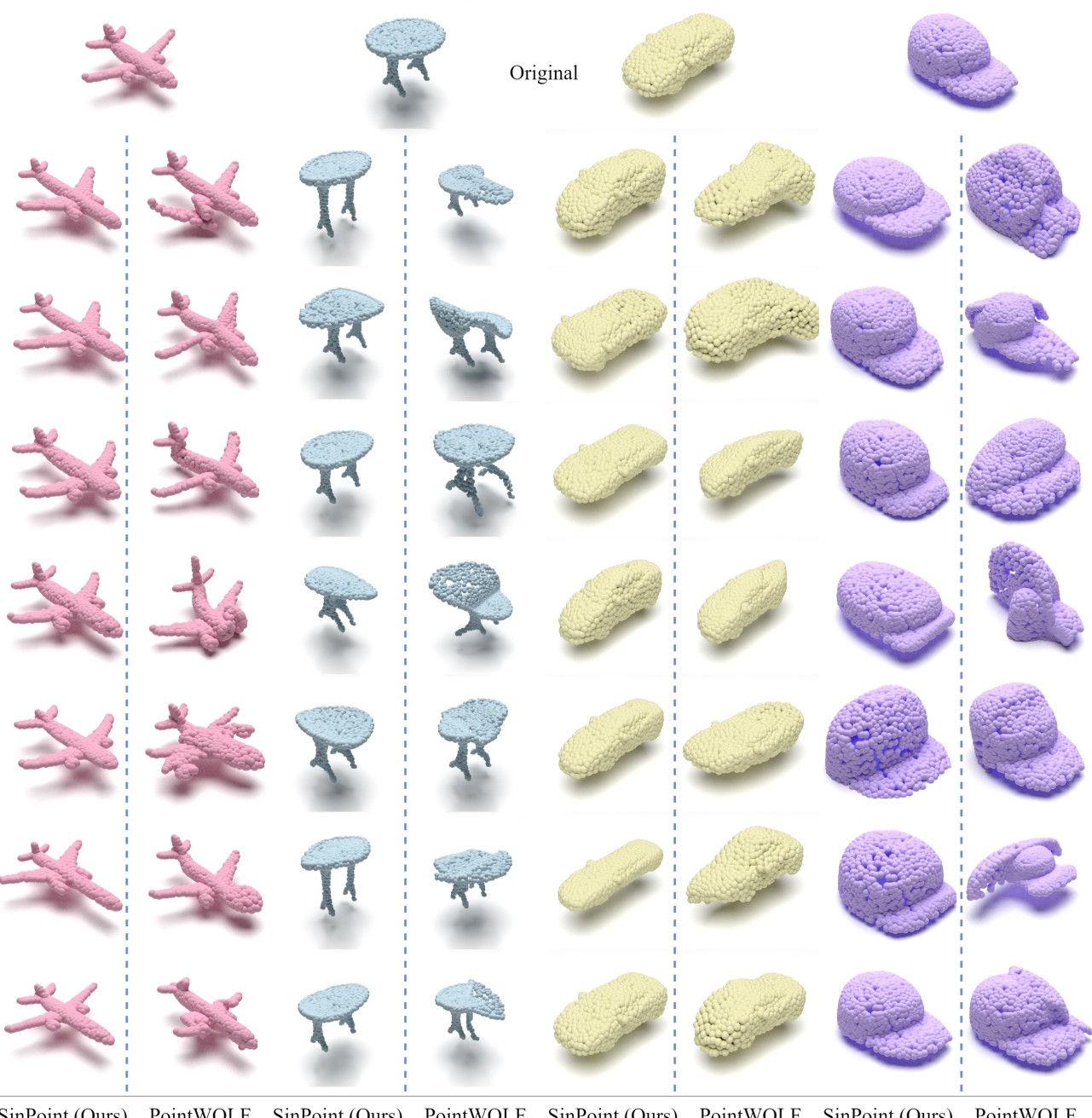

*Figure 11.* Visualization comparison of our SinPoint with PointWOLF. The top original point cloud, each shape left is SinPoint, right is PointWOLF. The samples generated by our SinPoint are more diverse and realistic.

