# OpenReview forum: "Rethinking Point Cloud Data Augmentation: Topologically Consistent Deformation"
_ICML.cc/2025/Conference — ICML 2025 poster_

### Official Review · Reviewer_Agtr · 2025-03-12

**Overall Recommendation:** 4

**Summary:**

This paper proposes a novel data augmentation method, SinPoint, for the 3d point cloud, leveraging the topological consistency deformation technique. It utilizes a sine-based mapping function for deformation under the Markov process. The approach has demonstrated the effectiveness of data augmentation by theory analysis and experiments.

**Claims And Evidence:**

see strength and weakness

**Essential References Not Discussed:**

No

**Experimental Designs Or Analyses:**

see strength and weakness

**Methods And Evaluation Criteria:**

see strength and weakness

**Other Comments Or Suggestions:**

none

**Other Strengths And Weaknesses:**

Strengths:
1.	This paper introduces homeomorphism, or the topological consistent structure, for data augmentation in point cloud analysis, which is a novel perspective.
2.	Combining the theoretical analysis and experiments, the paper demonstrates the importance of data expansions through data augmentation.
3.	The paper proposed SinPoint, which integrates the topological consistent sin-based deformation functions and the Markov chain. The results of 3D classification and part-segmentation have illustrated the effectiveness.


Weaknesses:
1.	Concepts of homeomorphism need further discussion. From mathematical aspects, the 3D surfaces in the mentioned datasets (e.g., ModelNet40) are mostly homeomorphisms across different classes. For example, the plane and guitar are homeomorphisms, while the plane and cup are not. The proposed method mostly employs the sine-based deform functions and designs the Markov chain data augmentation process, which is more likely to be a stable and continuous deformation process than a topology-consistent transformation.

**Questions For Authors:**

none

**Relation To Broader Scientific Literature:**

see strength and weakness

**Theoretical Claims:**

see strength and weakness

---

> ### Author Rebuttal · Authors · 2025-03-30
>
> Dear Reviewer Agtr:
>
> **Thanks for your time and insightful reviews. We appreciate your recognition of our work. We responded in detail as follows:**
>
> **Q1:** Concepts of homeomorphism need further discussion. From mathematical aspects, the 3D surfaces in the mentioned datasets (e.g., ModelNet40) are mostly homeomorphisms across different classes. For example, the plane and guitar are homeomorphisms, while the plane and cup are not.
>
> **A1:** Thank you for your recognition of our work. Indeed, as you say, when we look at different samples, their shapes and structures can be transformed into each other through a series of continuous deformations without involving holes or tears; then they are homeomorphic, such as a plane and a guitar. A cup with a handle has a hole, and the plane is non-hole, so they are not homeomorphic. **This is relative to the different samples.**
>
> **However, our SinPoint is to consider a single sample.** Our aim is **to make a single sample produce continuous smooth deformation through homeomorphic mapping,** which **makes the deformed sample maintain semantic consistency while increasing the geometric diversity of a single sample, so that the model can learn more discriminant features.**
>
> Data augmentation based on homeomorphism can enhance the diversity of a single sample, enabling the model to learn more discriminative features and **improve model performance (see Tables 1, 2, 6, 9, and 10). The anti-interference ability of the model is further improved (see Table 3).**
>
> **Since we are not specifically designing machine learning models to learn features of homeomorphisms between different samples, we do not consider homeomorphisms between different samples.**
>
> **Q2:** The proposed method mostly employs the sine-based deform functions and designs the Markov chain data augmentation process, which is more likely to be a stable and continuous deformation process than a topology-consistent transformation.
>
> **A2:** Thanks for your professional questions. Our data augmentation can be regarded as two processes, as shown in Figure 2. **First, a topology-consistent deformation sample is generated through sine transformation based on homeomorphism, and the original sample and the augmented sample are input into the Markov augmentation process based on the basic transformation to further increase the diversity of samples.**
>
> **From Table 4, we can find that the independent Markov gain is 1.0 and the independent SinPoint gain is 1.2. In this case, only augmented samples are used, and when further mixed samples are used to expand the variance of the data, the performance is improved again. In this case, the mixed Markov gain is 1.5, and the mixed SinPoint gain is 1.7. This also shows that SinPoint's performance improvement is always better than the Markov process. When the three work together, the performance gain is 2.0. This also shows that SinPoint and Markov can promote each other. And SinPoint played a crucial role.**
>
> **We thank you again for your careful review, which helped us a lot. We have addressed all your concerns in detail. If you have other questions, we can discuss them again, and we look forward to your feedback.**

---

> > ### Comment · Reviewer_Agtr · 2025-04-06
> >
> > Thanks for the rebuttal. My major concerns have been addressed well. Thus, I keep my rating.

---

> > > ### Author Response · Authors · 2025-04-07
> > >
> > > Dear Reviewer Agtr,
> > >
> > > Thanks again for recognizing our work.
> > >
> > > We appreciate the discussion and your positive feedback. Many thanks for your valuable time and effort.
> > >
> > > Best and sincere wishes,
> > >
> > > The authors

---

### Official Review · Reviewer_8z5w · 2025-03-13

**Overall Recommendation:** 3

**Summary:**

The paper proposes to use sine functions to augment the point cloud for the point cloud classification and segmentation tasks, with a Markov chain augmentation process to further improve the performance. The method achieves SOTA on different tasks with various backbones.

**Claims And Evidence:**

Although the method achieves SOTA performance, some of the claims are not well justified:

1. The paper's first main contribution is analyzing data augmentation from a statistical perspective, which is a strong claim. While I am not an expert in the theory of data augmentation, it's easy to find prior works related to this topic, for example [1]. Despite this claim, the paper’s analysis of data variance appears trivial, and it lacks an in-depth discussion on how increasing variance—specifically through the proposed augmentation—enhances model generalization to unseen data.

2. The second main contribution is proving that both a single sine function and a sum of multiple sine functions are homeomorphisms. However, this proof seems too trivial to be considered a significant contribution, i.e., given the definition, it's obvious that the functions are homeomorphisms.

3. While I acknowledge the novelty of the sine-based point cloud augmentation method, the contribution of the Markov chain augmentation process is not well justified. The process consists of four base transformations, none of which are novel. A better way to demonstrate its effectiveness would be to compare it against using each transformation individually (i.e., setting transition probabilities to 0) to isolate the impact of the Markov chain process itself.

[1] Dao et. al., 2019. A Kernel Theory of Modern Data Augmentation.

**Essential References Not Discussed:**

To the best of my knowledge all the essential related works are cited.

**Experimental Designs Or Analyses:**

1. The authors use SinPoint-SSF for classification and SinPoint-MSF for segmentation, which intuitively makes sense. However, it would be beneficial to show results where MSF is applied to classification and SSF to segmentation, as one could argue that MSF provides more diversity (higher variance), which might improve generalizability, as claimed in Section 3.

2. The Markov chain augmentation process and mix sampling were attached to SinPoint, leading to improved performance over baselines. In Table 4, mixing training samples appears to contribute significantly to the results. However, the authors did not specify the percentage of original samples used or explain how the mixup rate was chosen, which is important for reproducibility and understanding its impact.

3. The effectiveness of the Markov chain augmentation process still needs further justification, as I previously mentioned under Claims and Evidence.

**Methods And Evaluation Criteria:**

The proposed methods and/or evaluation criteria (e.g., benchmark datasets) make sense.

**Other Comments Or Suggestions:**

1. Definition 1 is not used anywhere in the paper, as it measures the variance of the prediction but not the data input.

2. Algorithm 1 doesn't include Markov chain augmentation process and sample mixing, this is confusing as it's named SinPoint.

**Other Strengths And Weaknesses:**

-

**Questions For Authors:**

I found the sine function-based augmentation interesting, and the results demonstrate its effectiveness. I would be happy to raise my score if the authors could address my concerns.

**Relation To Broader Scientific Literature:**

This method could be used in broader tasks in point-cloud understanding. It's also related to the theory of data augmentation.

**Theoretical Claims:**

The proofs are correct.

---

> ### Author Rebuttal · Authors · 2025-03-30
>
> Dear Reviewer 8z5w:
>
> **Thanks for your time and insightful reviews. We responded in detail as follows:**
>
> **Q1:** ...for example [1]. Despite this claim, the paper’s analysis ... appears trivial, and it lacks an in-depth discussion on how increasing variance—specifically through the proposed augmentation—enhances model generalization to unseen data.
>
> **A1:** Thank you for your valuable opinions, especially the one you mentioned [1], which is very valuable to us. This is a work that studies the basic theory of data augmentation, which provides deeper theoretical support for our work. Especially in the process of constructing Markov augmentation, our method is further proved to be correct and effective theoretically.
>
> Compared to [1] which mainly studies the theory of basic transformations, we propose a novel SinPoint to construct augmented samples. Meanwhile, this construction can steadily expand the variance of the data. As shown in Figure 2 (in https://github.com/Anonymous-code-share/ICML2025), the distribution of the augmented samples generated by our method is closer to the real distribution, which enables better coverage of the data distribution in the test set. This increased variance helps to reduce the bias of the model, thereby improving its ability to generalize to unseen data. Other methods can lead to huge biases that are detrimental to model learning. Due to word constraints, we will provide an in-depth analysis in the latest version.
>
> **Q2:** The contribution … sine functions are homeomorphisms. However, this proof seems too trivial to be …, … it's obvious ….
>
> **A2:** Thank you for your valuable advice. For a professional scholar like you, understanding is relatively simple. However, in the field of point cloud augmentation, most of the methods in Table 1 lack theoretical basis. Our method is the first one based on homeomorphic theory for point cloud augmentation, and we have to admit that it is indeed a significant contribution in the field of point cloud data augmentation. We hope that more detailed definitions and proofs will help different readers quickly understand the relevant theories of our SinPoint.
>
> **Q3:** A better way to demonstrate its effectiveness would be to compare it against using each transformation individually to isolate the impact of the Markov chain process itself.
>
> **A3:** Thank you for your constructive comments. We add ablation experiments to analyze the influence of different transformations on the model. It can be seen from the experiments that the Markov augmentation process has a better stable gain. The results are as follows:
>
> **Table R1: Ablation analysis for a single transformation with Markov.**
> | DGCNN | OA |
> |:---|:---:|
> | Markov | 93.7 |
> | scaling | 92.6 |
> | shifting | 92.1 |
> | rotation | 92.9 |
> | jittering | 92.7 |
>
> **Q4:** However, it would be beneficial to show results where MSF is applied to classification and SSF to segmentation. (Justification for two methods (SSF and MSF))
>
> **A4:** Thank you and Reviewer SdQc for your constructive comments. This experimental analysis was also considered in an earlier manuscript but was removed due to formatting issues in the submission version. According to your suggestion, we will re-add this part of the experiment and analysis in the latest version, and some results are compared as follows:
>
> **Table R2: SSF and MSF on classification and segmentation.**
> | DGCNN | Classification (OA) | Segmentation (mIoU) |
> |:---|:---:|:---:|
> | SinPoint-SSF | **90.2** | 85.3 |
> | SinPoint-MSF | 89.8 | **85.5** |
>
> **Q5:** However, the authors did not specify the percentage of original samples used or explain how the mixup rate was chosen, which is important for reproducibility and understanding its impact.
>
> **A5:** Thanks for your careful review. We show in Figure 5 that the original sample and the enhanced sample are input together for training, without setting the mixing rate. As can be seen from Table 4, SinPoint alone can improve the performance by 1.2, and the improvement after mixing samples is only 0.5. Mixed samples do not play a dominant role in performance improvement. SinPoint and the Markov augmentation process are the decisive factors.
>
> **Q6:** Definition 1 is not used anywhere in the paper, as it measures the variance of the prediction but not the data input.
>
> **A6:** We put Definition 1 in the text to facilitate logical coherence and reader understanding because Definition 1 and Theorem 1 are related. And definition 1 is also used in the proof of Theorem 1 in the supplementary material.
>
> **Q7:** Algorithm 1 is confusing as it's named SinPoint.
>
> **A7:** Thanks for reminding me. We will modify the title of algorithm 1 to SinPoint without Markov. The pseudo-code of the Markov process is added in the latest version.
>
> **We thank you again for your careful review, which helped us a lot. We have addressed all your concerns in detail. If you have other questions, we can discuss them again, and we look forward to your feedback.**

---

> > ### Comment · Reviewer_8z5w · 2025-04-09
> >
> > Thank you for the detailed rebuttal. I still believe the theoretical analysis in this paper is relatively weak and would benefit from a more in-depth discussion. However, considering the novelty of the sine-based point cloud augmentation method, the SOTA performance, and the additional ablation studies demonstrating the method’s effectiveness, I would raise my score to weak accept.

---

> > > ### Author Response · Authors · 2025-04-09
> > >
> > > Dear Reviewer 8z5w,
> > >
> > > Thanks again for recognizing our work.
> > >
> > > We appreciate the discussion and your positive feedback. Many thanks for your valuable time and effort.
> > >
> > > We promise to add more in-depth discussion and theoretical analysis in the latest version.
> > >
> > > Best and sincere wishes,
> > >
> > > The authors

---

### Official Review · Reviewer_SdQc · 2025-03-14

**Overall Recommendation:** 2

**Summary:**

This paper presents "SinPoint," a new data augmentation method for point clouds. The main idea is to deform point clouds in a way that's supposed to preserve their overall structure, using sine functions to create the deformations. The authors argue, using the concept of homeomorphism, that these deformations don't change the underlying topology of the point cloud.

They offer two versions of their method: SinPoint-SSF, which uses a single sine wave and is suggested for classification, and SinPoint-MSF, which combines multiple sine waves and is proposed for segmentation. They also describe a way to combine SinPoint with other common transformations like rotation and scaling, using a Markov Chain process to create more varied augmented data.

The paper's main results show that SinPoint generally performs better than other point cloud augmentation methods on several datasets (ModelNet40, ReducedMN40, ScanObjectNN, and ShapeNetPart) and with a few different network architectures (PointNet, PointNet++, DGCNN). They report improvements in accuracy for classification and IoU for segmentation. The paper also claims SinPoint makes models more robust to things like noise and rotation. The contributions seem to be the sine-based deformation, the homeomorphism argument, the Markov Chain process, and the experimental results. The authors support these claims with experiments, ablation studies, and some visualizations.

## Update after rebuttal
I have carefully read the authors' detailed rebuttals (both the initial response and the reply to my comment) and the comments from other reviewers. I appreciate the authors' engagement and clarifications.

The authors pointed out that results on complex scene datasets (S3DIS and SemanticKITTI) were included in Table 10 of the supplementary material, addressing one of my primary concerns about the lack of such evaluation. They also provided further rationale for the choice of the sine function and addressed concerns about computational cost by referencing Table 11.

However, my core assessment remains largely unchanged, and I will keep my current rating. My main concern is not the absence of these results, but their limited prominence and analysis within the main paper. Evaluating performance on complex, realistic data is crucial for convincingly demonstrating the significance and generalizability of a data augmentation method for point clouds. The main paper's experimental narrative heavily relies on the simpler datasets, which limits the strength of the evidence presented for broad applicability.

Therefore, while acknowledging the authors' responses and the additional data provided, the way the experimental validation is presented in the main body of the work still leaves questions about the method's practical impact in more challenging, real-world scenarios.

**Claims And Evidence:**

The paper makes several key claims regarding SinPoint's effectiveness. The first is that SinPoint generates topologically consistent deformations. This is supported by a theoretical proof (Theorem 2 and Appendix C) demonstrating that the sine-based transformation is, under specific conditions, a homeomorphism. While the mathematical reasoning is sound, the practical implications might be a bit overstated. Real-world point clouds are discrete and can be quite complex. It's possible that large deformations, even if theoretically homeomorphic, could still distort the shape in a way that effectively changes the topology from a machine learning perspective.

Another central claim is that SinPoint outperforms existing point cloud data augmentation methods. The paper provides quantitative results in Tables 1 and 2, showing improvements in accuracy and mIoU across several datasets and network architectures. While the results generally favor SinPoint, the improvements are sometimes relatively modest. A much bigger concern is that the experiments are limited to relatively simple point cloud datasets. Without results on complex (PartObjaverse-Tiny), scene-level data (like S3DIS, ScanNet, or SemanticKITTI), it's hard to know how well the proposed method truly generalizes.

The paper also claims that SinPoint makes models more robust to corruptions, supported by the results in Table 3. While this is a positive finding, it shares the same limitation as the performance claim: robustness on simpler datasets doesn't guarantee robustness in complex, real-world scenarios. The contribution of the Markov Chain Augmentation Process is demonstrated in the ablation studies (Table 4), but the benefit seems incremental. One might question whether the added complexity is worth the relatively small gains. The justification for selecting sine function lacks compelling evidence.

Overall, while the theoretical claim of topological consistency is mathematically strong, the core claim of superior performance is weakened by experimental limitations. The robustness claim has similar limitations. The benefit of the Markov Chain is supported, but its impact seems modest.

**Essential References Not Discussed:**

I did not find essential references that were missing from the paper's discussion.

**Experimental Designs Or Analyses:**

**Strengths:**
- The inclusion of robustness experiments (Table 3) is a good point.
- The experiments use a variety of datasets and backbone networks.
- The ablation studies (Table 4) seem carefully designed and help to isolate the contribution of different components of the proposed method (SinPoint, Markov Chain, mixed training).

**Weaknesses:**
- The experiments are primarily conducted on relatively simple datasets. ModelNet40 consists of clean CAD models, ScanObjectNN features isolated objects, and ShapeNetPart deals with individual object parts. This limits the conclusions that can be drawn about the proposed method's effectiveness in real-world, complex scenarios.
- More qualitative analysis, especially showing examples of failure cases or situations where the method struggles, would provide a more balanced view.

**Methods And Evaluation Criteria:**

The proposed method, SinPoint, has some conceptually appealing aspects, but there are also some questions about whether it fully makes sense for the broader problem of point cloud data augmentation, particularly in real-world applications.

The use of a sine function to generate deformations is, on the surface, a reasonable idea. It allows for smooth, controlled deformations, and the connection to homeomorphism provides a nice theoretical grounding. However, the paper doesn't convincingly justify why a sine function is the best choice, compared to other possible deformation methods. There's also a lack of comparison to other techniques from computer graphics, like thin-plate splines or free-form deformation.

The two variants, SinPoint-SSF and SinPoint-MSF, are presented as being suitable for classification and segmentation, respectively. This makes intuitive sense, as segmentation often requires more localized deformations. The Markov Chain Augmentation Process, while adding flexibility, also adds complexity, and it's not entirely clear if the benefits outweigh the costs.

The choice of evaluation criteria, specifically the datasets, is a significant area of concern. While ModelNet40, ReducedMN40, ScanObjectNN, and ShapeNetPart are standard benchmarks, they represent relatively simple point clouds. They don't capture the complexity of many real-world applications (occlusions, clutter, noise, varying point density).

In summary, while the core idea of SinPoint has merit, the lack of comparison to alternative deformation methods and, crucially, the limited evaluation on complex, scene-level datasets, raise questions about whether the proposed methods and evaluation criteria are fully adequate.

**Other Comments Or Suggestions:**

The authors could consider making the code publicly available upon publication.

**Other Strengths And Weaknesses:**

**Other Strengths:**
- The paper is generally well-written and easy to follow. The core concepts are explained clearly. The use of formal definitions and theorems adds to the technical clarity.

**Other Weaknesses:**
- Justification for two methods (SSF and MSF): It would be useful if the author could provide more evidence on this.

**Questions For Authors:**

Please refer to the previous sections.

**Relation To Broader Scientific Literature:**

The paper cites relevant work in point cloud processing and data augmentation. It also acknowledges the general success of data augmentation in computer vision. However, the paper could do a better job of connecting to the broader scientific literature, particularly in computer graphics. While the paper uses the concept of homeomorphism, citing a relevant topology textbook, it misses a key connection to the extensive research on non-rigid deformation in computer graphics.

**Theoretical Claims:**

The paper presents two main theoretical claims, both related to homeomorphism:

- Theorem 1 (Data augmentation increases the variance of the dataset): This theorem seems correct and its proof in Appendix C is sound. The derivation is straightforward.

- Theorem 2 (Homeomorphism Based on Sine Function): In Appendix C. The reasoning also seems correct.

---

> ### Author Rebuttal · Authors · 2025-03-30
>
> Dear Reviewer SdQc:
>
> **Thanks for your time and insightful reviews. We responded in detail as follows:**
>
> **Q1:** It's possible that large deformations, even if theoretically homeomorphic, could still distort the shape in a way that effectively changes the topology.
>
> **A1:** Thank you for agreeing with our theory. Homeomorphism is strictly maintained in theory and has been proved by theorem 2. **If the topology is inconsistent, holes or tears will be created after deformation, but our continuous deformation based on sine transformations does not create holes or tears and therefore does not change the topology.** However, in machine learning, **excessive deformation can lead to ambiguity,** which we need to avoid. As we have verified in Table 3, too large deformation parameters will be detrimental to model learning. Therefore, we choose the appropriate parameters to generate the enhanced sample.
>
> **Q2:** A much bigger concern is … limited to … simple … datasets. Without … on complex … datasets.
>
> **A2:** Your question is very professional. In terms of experiments, **we also tested on complex datasets, and Table 10 in the supplementary material shows our experimental results on S3DIS and SemanticKITTI.**
>
> **Q3:** The paper doesn't justify why a sine function is the best choice, compared to other possible deformation methods. a lack of comparison to …, like thin-plate splines or free-form deformation.
>
> **A3:** Your questions are very professional and very helpful to us. **We clarify that we are not saying that the sine function is the best choice.** We discussed why the sine function was chosen, **and we have a brief introduction in Section 4.3.** The reasons are as follows: **The sine function is a periodic function, and by controlling the amplitude, the maximum deformation can be easily adjusted. Other functions, such as monotonic functions, parabolic functions, or other types of functions, cause excessive deformation as the diameter of the object increases, which leads to augmented samples deviating from the real ones.**
>
> In contrast, thin-plate splines are computationally more complex and may not be suitable for current applications. Free-form deformation (FFD) can provide flexible deformation control, but the process is complicated. Thus, it is simpler and more efficient to use the sine function directly.
>
> This can be our future exploration direction because FFD can generate more controlled deformation samples. **We would like to thank the reviewers for their valuable comments, which also guide us to a new exploration direction.**
>
> **Q4:** The Markov Chain Augmentation Process, while adding flexibility, also adds complexity, and it's not entirely clear if the benefits outweigh the costs.
>
> **A4:** Thank you for affirming our method. **We have given a quantitative analysis of your concerns in Table 11 (in supplementary material). Compared to existing methods, such as PointAugment, our method reduces time costs by up to 10 times and improves performance by 11%. We not only save the calculation cost but also improve the accuracy of the model.**
>
> **Q5:** The choice of evaluation criteria, specifically the datasets.
>
> **A5:** Thanks for your suggestion. To make a fair comparison with other methods, we had to conduct primary experiments on these datasets. **However, we have also carried out additional experiments, including S3DIS and SemanticKITTI, in Table 10 of supplementary materials.** Moreover, in Table 3, we further investigate the robustness of the SinPoint by simulating complex noises in the real world. We plan to explore SinPoint's performance on more realistic and complex datasets in future work.
>
> **Q6:** More qualitative analysis, ... showing examples of failure cases
>
> **A6:** Thanks for reminding me. **We added it in the latest version, and you can see more qualitative analysis examples of failure cases in Figure 1** (in https://github.com/Anonymous-code-share/ICML2025). Meanwhile, we also conducted ablation experiments as shown in Table 7. When excessive deformation occurs, the performance will decline.
>
> **Q7:** It misses a key connection to the extensive research on non-rigid deformation in computer graphics.
>
> **A7:** Your question is good, **especially your proposed FFD method for computer graphics, which encourages us to develop controlled deformation augmentation in the future. We plan to discuss the feasibility of these methods (like TPS or FFD) as part of our related work.**
>
> **Q8:** Justification for two methods (SSF and MSF).
>
> **A8:** Due to the word limit, we jointly replied to the same questions in **Q4 of Reviewer 8z5w.**
>
> **Q9:** code publicly available upon publication.
>
> **A9:** Yes, we must do it.
>
> **We thank you again for your careful review, which helped us a lot. We have addressed all your concerns in detail. If you have other questions, we can discuss them again, and we look forward to your feedback.**

---

> > ### Comment · Reviewer_SdQc · 2025-04-05
> >
> > Thank you to the authors for providing a detailed rebuttal and addressing the questions raised. I have read the rebuttal carefully, along with the comments from the other reviewers.
> > While I appreciate the clarifications and the additional results pointed out in the supplementary material, my core concerns about the extent and prominence of the evaluation on complex, real-world scenarios remain.
> > Therefore, I will maintain my current rating.

---

> > > ### Author Response · Authors · 2025-04-06
> > >
> > > Dear Reviewer SdQc:
> > >
> > > **We appreciate your response. We are glad to have addressed almost all of your concerns. We have once again concluded our paper, and we hope that the following reply will truly help you.**
> > >
> > > First, **we would like to clarify that our SinPoint is applied to object point cloud tasks, not scene point clouds (in line 95).** However, the applicability of our SinPoint to complex scene point cloud tasks has already been tested **in Table 10. Our method shows a performance improvement of 0.6 on S3DIS and up to 7.6 on SemanticKITTI, which is sufficient to prove the advanced nature and scalability of our method. In fact, we have also addressed this in Q2 of the rebuttal.**
> > >
> > > **Object and scene point cloud tasks are two different research directions.** Object tasks help computers understand objects in the 3D world more accurately. The main goal of scene tasks is to understand and analyze the entire 3D scene, including the objects in the scene, their structures, and the spatial relationships between them. **The paper, "Advancements in Point Cloud Data Augmentation for Deep Learning: A Survey, PR2024," is a review of point cloud augmentation, which classifies and summarizes the tasks and applications of point clouds and introduces the scope of application of different cloud augmentation methods.**
> > >
> > > **Our main research is object point cloud augmentation to improve the model's understanding of objects.** The existing point cloud augmentation methods may improve data diversity to some extent, but they often overlook the point cloud's intrinsic structure and semantic details, resulting in a loss of topological consistency in the augmented point cloud. For instance, PointMixup, PointCutMix, and SageMix all use different strategies to mix samples, but they do not consider the local structure of each sample. PointAugment relies on a learnable transformation matrix, making the outcome unpredictable. Similarly, PointWOLF transforms local point clouds using a combination of strategies, which can lead to data distortion and significant semantic deviation, as shown in Figure 1.
> > >
> > > **Thus, we mainly solve the problem of topological inconsistencies in the current object point cloud augmentation methods.**
> > >
> > > **Our contributions are:**
> > >
> > > We analyze the data augmentation from a statistical perspective. This expands the distribution boundary of the dataset and increases its variance.
> > >
> > > We prove that the proposed sine-based mapping function is a homeomorphism. In theory, it increases the diversity of point clouds without destroying the topological structure.
> > >
> > > We propose a new Markov chain augmentation framework that increases sample diversity by randomly combining different foundational transformations to expand the distribution space of the dataset.
> > >
> > > **We demonstrate the effectiveness of our framework by showing consistent improvements over SOTA augmentation methods on both synthetic and real-world datasets in 3D shape classification and part segmentation tasks.**
> > >
> > > **Datasets used by different-level tasks:**
> > >
> > > The main datasets used in object-level tasks include 1) Synthetic datasets for classification tasks: ModelNet10 (MN10), ModelNet40 (MN40), ReducedMN40 (RMN40). 2) Real-world scanned datasets for classification tasks: ScanObjectNN (SONN). 3) Synthetic datasets for part classification tasks: ShapeNetPart (SNP).
> > >
> > > The main datasets used in scene-level tasks include 1) Indoor segmentation task dataset: S3DIS. 2) Outdoor segmentation task dataset: SemanticKITTI.
> > >
> > > **We focus on object point cloud tasks, and the methods we compare are also focused on object-level tasks. We summarize our object-level method as follows:**
> > >
> > > Table R: The datasets used in the object-level point cloud augmentation.
> > > |Method|Object-level|Scene-level|
> > > |:---|:---:|:---:|
> > > | PointAugment(CVPR2020)|MN10, MN40|NA|
> > > | PointMixup(ECCV2020)|MN40, SONN|NA|
> > > | PointWOLF(ICCV2021)|MN40, RMN40, SONN, SNP|NA|
> > > | RSMix(CVPR2021)|MN10, MN40|NA|
> > > | PatchAugment(ICCVW2021)|MN10, MN40, SONN|NA|
> > > | SageMix(NeurIPS2022)|MN40, SONN|NA|
> > > | WOLFMix(PMLR 2022)|MN40|NA|
> > > | PCSalMix(ICASSP 2023)|MN10, MN40|NA|
> > > | PointPatchMix(AAAI2024)|MN40, SONN|NA|
> > > | SinPoint(Ours)|MN40, RMN40, SONN, SNP|S3DIS, SemanticKITTI|
> > >
> > > As shown in the table above, **in object-level tasks, other methods are tested only on object-level datasets, and we use the most object-level datasets to verify the performance of our methods.** However, **to further expand the application scope of our method, we also conducted additional validation on complex scene point clouds, such as S3DIS and SemanticKITTI. Compared to object-level methods, we are already SOTA. And on scene-level tasks, we are also able to further expand and enhance performance.** In the future, **our method needs to add new contributions to better apply to scene tasks, which is our future work.**
> > >
> > > **We hope that our sincere response will help you and other reviewers understand our research area. We look forward to your feedback.**

---

### Official Review · Reviewer_MuAY · 2025-03-14

**Overall Recommendation:** 3

**Summary:**

The paper introduces SinPoint, a novel data augmentation technique for point clouds that employs homomorphism-based sine transformations to increase geometric diversity while preserving topological consistency. SinPoint has two variants: SinPoint-SSF, which uses a single sine function anchored at the origin, and SinPoint-MSF, which combines multiple sine functions anchored at different points to produce richer deformations. Experiments demonstrate that SinPoint consistently outperforms state-of-the-art augmentation methods, enhancing the generalization and robustness of models (PointNet, PointNet++, DGCNN) on synthetic and real-world datasets across 3D shape classification and part segmentation tasks.

## update after rebuttal
Due to the limit of only one round of questions and answers between the reviewer and the authors, therefore I still have some minor concerns below. However, these concerns will not strongly affect my final rating of **Weak Accept** (leaning towards accept).

Q1: Can we use the same parameters for all datasets, or do we need to choose different values for each dataset? Are there any special requirements for our own collected data?

Q2 and Q3: To truly validate effectiveness, data augmentation methods should perform well across both old and new, simple and complex models. Relying on simpler models, which are often less representative of current state-of-the-art systems, may fail to capture real-world challenges like scalability, robustness, or compatibility with modern architectures. A stronger approach would test a broad spectrum of simple to advanced models to ensure versatility, rather than tuning methods to succeed only on legacy frameworks. While starting with smaller models is practical, it's flawed to assume their performance predicts success on large models, which can exhibit unique behaviors and sensitivities not seen in simpler systems.

Q5: Could the authors provide more details on the training efficiency of the proposed method? For example, the OA of the original method, and then when applying SinPoint, the training time of the original method, and then when applying SinPoint. Some of the above methods can be used: PointNet, PointMetaBase, SPoTr, and Point Transformer v3.

**Claims And Evidence:**

The claims are clear and supported by evidences.

**Essential References Not Discussed:**

None

**Experimental Designs Or Analyses:**

Please check the Methods And Evaluation Criteria above.

**Methods And Evaluation Criteria:**

- While homeomorphism theoretically offers diverse continuous transformations for data augmentation, it can distort critical geometric features such as distances, angles, and curvature. As evident in Figure 2 in the main paper and Figures 8 and 9 in the supplementary, such transformations may generate unrealistic data.
- In addition, when dealing with more complicated task (part segmentation task as we can see on Table 2 in the main paper or Table 9 in the supplementary), the proposed method’s performances dropped significantly (less than 1%, for some methods they are just 0.1 or 0.2%, compared to 2.6-7.3% on classification task), it dues to confusing part boundaries (mixed up the point-wise labels) which leads to incorrect training signals. Even though the authors tried to show in Table 10 that their method can work well in scene segmentation (with SemanticKITTI, a LiDAR, and a sparse dataset, which can get more information from any data augmentation methods), the chosen method, MinkNet, is pretty old (2019).
- Moreover, the experimental evaluation has limitations: the proposed method shows effectiveness with older augmentation techniques (Table 1) but is less effective when combined with recent methods (Table 6).

**Other Comments Or Suggestions:**

None.

**Other Strengths And Weaknesses:**

The paper is well-structured, making it accessible and understandable.

**Questions For Authors:**

1. Could the authors provide a detailed analysis of how the proposed method specifically impacts part boundaries and label consistency?
2. The authors should report additional details on model size, number of parameters, and throughput when integrating the proposed augmentation method with different backbone models.

**Relation To Broader Scientific Literature:**

It somehow has impact to the scientific literature on a specific domain, which is Data Augmentation for Point Cloud Understanding.

**Theoretical Claims:**

The proofs are correct for theoretical claims.

---

> ### Author Rebuttal · Authors · 2025-03-30
>
> Dear Reviewer MuAY:
>
> **Thanks for your time and insightful reviews. We responded in detail as follows:**
>
> **Q1:** ...such transformations may generate unrealistic data.
>
> **A1:** Your opinion is very professional. Indeed, when the deformation parameter is too large, it will produce unrealistic data. Therefore, **we chose smaller parameters to avoid such results, and ablation experiments were also conducted in Table 7.** When the deformation is too large, it will also reduce the performance of the model. **Our SinPoint aim is to increase the geometric diversity of the augmented samples while maintaining topological consistency to understand the nature of the data from more viewpoints and conditions, which can effectively avoid the overfitting of the model to the training data.**
>
> **Q2:** ...it dues to confusing part boundaries ... which leads to incorrect training signals. Even though the authors tried to show in Table 10 that their method can work well in scene segmentation ..., the chosen method, MinkNet, is pretty old (2019).
>
> **A2:** Your observations were sharp and your reviews were professional. **To avoid boundary confusion, we introduce more conservative parameters to ensure that the transformed part boundaries remain as accurate as possible and avoid extreme distortions. In addition, as can be seen from Table 7, the appropriate deformation can maximize model performance.**
>
> In the experiment, we also found that the performance improvement in the part segmentation was not significant and conducted a series of experimental verifications, as shown in Table 9 and Table 10. **We find that most of the data augmentation methods in Table 1 are not suitable for segmentation tasks. As shown in Table 2, only four works reported the result of part segmentation. But our SinPoint is best.**
>
> However, to confirm whether the method has limitations, we further verify it on multiple backbones, as shown in Table 9, and it can be found that our method can achieve certain performance improvement.
>
> **Further, we test on scene segmentation; in Table 10, the performance improvement is still limited when performing semantic segmentation of S3DIS, while the performance improvement is significant when performing instance segmentation on SemanticKITTI.**
>
> We analyzed that ShapeNetPart and S3DIS are fine-grained semantic segmentation datasets, which are more challenging, while SemanticKITTI is a coarse-grained instance segmentation dataset. As shown in Table 1, our method is better suited for object tasks, which explains its superior performance on SemanticKITTI. Our main goal was to **use MinkNet to test the limitations of our approach to segmentation tasks, which has been demonstrated in experiments. So we didn't add more scene-level backbone to test.**
>
> **Q3:** the proposed method shows effectiveness with older ... (Table 1) but is less effective ... with recent methods (Table 6).
>
> **A3:** Your question is valuable. **This issue is largely influenced by the dataset and model capabilities. The older backbone design is simple and prone to overfitting due to insufficient data, and data augmentation is to avoid this problem by increasing the diversity of data.** Therefore, data augmentation can greatly improve its performance. **However, in the recent backbone, due to its more powerful network design and strong representation ability, the backbone is close to the theoretical limit, and the marginal revenue is reduced.**
>
> **Therefore, all current point cloud augmentation efforts are based on the old backbone to test the performance of the method, which can highlight the advantages of these methods.**
>
> **Q4:** Detailed analysis the impacts part boundaries and label consistency
>
> **A4:** Your question is very professional and valuable. **We have carried out relevant analysis in the first draft, but this part has been deleted from the submission version due to the structural adjustment.** For the boundary analysis, **we deeply discuss the relationship between the determinant of the Jacobian and the parameter of sine transformation.** Regarding label consistency, **we analyze it from the property of one-to-one mapping of homeomorphism.** Due to the rebuttal **word limit**, **we will re-add these in the latest version.**
>
> **Q5:** The authors should report additional details on model size, number of parameters, and throughput.
>
> **A5:** Our SinPoint is **a plug-and-play, non-parameter method and is only used during the training phase. It is removed during the testing phase.** Therefore, **our SinPoint does not change the size, number of parameters, and throughput of the baseline.** However, **to show them more clearly to readers in the future, we plan to report these details and add them to the latest version.**
>
> **We thank you again for your careful review, which helped us a lot. We have addressed all your concerns in detail. If you have other questions, we can discuss them again, and we look forward to your feedback.**

---

> > ### Comment · Reviewer_MuAY · 2025-04-01
> >
> > Thank you to the authors for their response and efforts to address my concerns. While I appreciate the attempt, several points remain inadequately resolved, though the authors have acknowledged some of my comments.
> >
> > Q1: The decision to select smaller parameters to avoid unrealistic data, which results in a performance drop, does not constitute a meaningful scientific contribution. This approach resembles parameter fine-tuning for optimal performance rather than a novel advancement.
> > - Q2 and Q4: The responses regarding the impact of part boundaries and label consistency are incomplete. However, the authors have committed to addressing these in the final version, which I appreciate.
> > - Q2 and Q3: The rationale for relying on older methods lacks robustness. A critical perspective might question, "Why propose a new method to enhance outdated techniques instead of leveraging recent approaches with advanced network designs and superior representation capabilities?"
> > - Q5: Similar concerns apply here, though the authors have again promised to incorporate improvements in the final version.
> >
> > Given these considerations, I maintain my rating of "Weak Accept". I encourage the authors to fulfill their commitment by including the promised updates in the final manuscript.

---

> > > ### Author Response · Authors · 2025-04-02
> > >
> > > Dear Reviewer MuAY:
> > >
> > > **Thanks for your time and reply. We hope the following responses truly address your concerns.**
> > >
> > > **Q1:** The decision to select smaller parameters to avoid unrealistic data, which results in a performance drop, does not constitute a meaningful scientific contribution. This approach resembles parameter fine-tuning for optimal performance rather than a novel advancement.
> > >
> > > **A1:** Thanks for your reply and further concern about the issue. Our detailed reply is as follows:
> > >
> > > First, **we are not fine-tuning the sine function parameters. As shown in Algorithm 1 (line 220), we generate diverse deformation functions by uniformly sampling from different parameter ranges. Specifically, in Table 7, parameter $A$ refers to a parameter range of $[-A, A]$, and the amplitude is uniformly sampled from this range, not fixed.**
> > >
> > > Second, **the ablation of the parameter range is used to validate the upper and lower bounds of the method, not for fine-tuning the sine function parameters.** In Table 7, **we can observe that our method remains SOTA across different parameter ranges, which further illustrates that the robustness of our SinPoint comes from the SinPoint itself.**
> > >
> > > Finally, **we are the first to propose a novel method based on homeomorphism and a Markov augmentation process, which provides theoretical guarantees for data augmentation. This further demonstrates the novelty and effectiveness of our method.**
> > >
> > > **Q2 and Q4:** The responses regarding the impact of part boundaries and label consistency are incomplete. However, the authors have committed to addressing these in the final version, which I appreciate.
> > >
> > > **A2 and A4:** Thanks for your responses. **Detailed analysis of label consistency in line 706 of our paper.** Due to the word limit, we commit to providing a detailed analysis of part boundaries in the final version. Part of the analysis about part boundaries is as follows:
> > >
> > > **The Jacobian determinant is a quantity that describes the 'stretching' or 'shrinking' properties of the mapping locally. In practical applications, when the sign of the Jacobian determinant changes, a transition from 'stretching' to 'shrinking' may occur, leading to a folding phenomenon.**
> > >
> > > Given the mapping $P' = P + Asin(\omega P + \phi)$, we can see it as a transformation from $P$ to $P'$, where $P =(x,y,z)$ and $P' =(x',y',z')$ denote points in 3D space.
> > >
> > > The determinant of this Jacobian is: $det(J_{P'}(P)) = (1+A_{x}\omega_{x}cos(\omega_{x}x + \phi_{x})) \cdot (1+A_{y}\omega_{y}cos(\omega_{y}y + \phi_{y})) \cdot (1+A_{z}\omega_{z}cos(\omega_{z}z + \phi_{z})).$
> > >
> > > **A larger $A$ increases the probability that the Jacobian determinant becomes zero, making the transformation more intense and causing stronger folding of the shape and structure. This is consistent with our visualization, as shown in Figure 1 (in https://github.com/Anonymous-code-share/ICML2025), where folding only occurs when $A$ is large.**
> > >
> > > **Q2 and Q3:** The rationale for relying on older methods lacks robustness. A critical perspective might question, "Why propose a new method to enhance outdated techniques instead of leveraging recent approaches with advanced network designs and superior representation capabilities?"
> > >
> > > **A2 and A3:** Thanks for your further discussion. Your question is very critical.
> > >
> > > Firstly, **we are not enhancing outdated technologies, but rather using these technologies to validate the effectiveness of our data augmentation methods.** Data augmentation methods are generally validated for effectiveness on simple models, such as PointAugment (CVPR 2020), PointWOLF (ICCV 2022), and PointPatchMix (AAAI 2024).
> > >
> > > Secondly, data augmentation is typically applied in contrastive learning or pretraining large models to construct positive samples. Due to the difficulty of training large models, we cannot directly validate the performance of new data augmentation methods on large models. Therefore, after validating the effectiveness of the methods on smaller models, they are directly applied to large model training. **Specifically, data augmentation plays a crucial role in contrastive learning models like CLIP (Contrastive Language-Image Pretraining) and BLIP.**
> > >
> > > **Q5:** Similar concerns apply here, though the authors have again promised to incorporate improvements in the final version.
> > >
> > > **A5:** We promise to add these model parameters in the final version. We have updated the table as follows:
> > >
> > > Table R2: Additional details on backbones.
> > > |Backbone|OA|Params. (M)|FLOPs (G)|Throughput (ins./sec.)|
> > > |:---|:---:|:---:|:---:|:---:|
> > > |PointNet|70.8|3.5|0.9|4212|
> > > |PointNet++|84.5|1.5|1.7|1872|
> > > |DGCNN|84.6|1.8|4.8|402|
> > > |PointMLP|87.5|13.2|31.3|191|
> > > |PointNeXt-S|88.9|1.4|1.6|2040|
> > > |PointMetaBase-S|89.3|1.4|0.6|2674|
> > > |SPoTr|89.5|1.7|10.8|281|
> > >
> > > **We thank you again for your careful review. We have addressed all your concerns in detail. If you have other questions, we can discuss them again, and we look forward to your feedback.**

---

### Decision · Program_Chairs · 2025-05-01

**Decision:**

Accept (poster)

**Comment:**

This paper proposes a novel data augmentation technique for point clouds called SinPoint. It leverages homomorphism-based sine transformations to increase geometric diversity while preserving topological consistency. The manuscript received mixed reviews with one weak reject, two weak accept, and one accept.

The main concern centers on insufficient prominence and analysis within the main paper.
However, AC acknowledges that these experiments and analyses can be added in the final version to address the reviewer's concerns.
After carefully evaluating the paper, the authors' responses, and the reviewers' comments, AC decided to accept it.
The authors must follow through on their commitments and substantially revise the paper to include the promised experiments and analyses.